# Affect-congruent attention modulates generalized reward expectations

**Daniel Bennett**[1]⊙*, **Angela Radulescu**[2]⊙*, **Sam Zorowitz**[3], **Valkyrie Felso**[4], **Yael Niv**[3,5]

**1** School of Psychological Sciences, Monash University, Clayton, Australia, **2** Department of Psychiatry, Icahn School of Medicine at Mount Sinai, New York, New York, United States of America, **3** Princeton Neuroscience Institute, Princeton University, Princeton, New Jersey, United States of America, **4** Max Planck Institute for Intelligent Systems, Tübingen, Germany, **5** Department of Psychology, Princeton University, Princeton, New Jersey, United States of America

⊙ These authors contributed equally to this work.
* daniel.bennett@monash.edu (DB); angela.radulescu@mssm.edu (AR)

**Data Availability Statement:** Data and code underlying analyses reported in this manuscript are publicly available at https://osf.io/egw5c/. Raw output files were analysed using NivLink, an open-source Python package for preprocessing EyeLink

## Abstract

Positive and negative affective states are respectively associated with optimistic and pessimistic expectations regarding future reward. One mechanism that might underlie these affect-related expectation biases is attention to positive- versus negative-valence features (e.g., attending to the positive reviews of a restaurant versus its expensive price). Here we tested the effects of experimentally induced positive and negative affect on feature-based attention in 120 participants completing a compound-generalization task with eye-tracking. We found that participants' reward expectations for novel compound stimuli were modulated in an affect-congruent way: positive affect induction increased reward expectations for compounds, whereas negative affect induction decreased reward expectations. Computational modelling and eye-tracking analyses each revealed that these effects were driven by affect-congruent changes in participants' allocation of attention to high- versus low-value features of compounds. These results provide mechanistic insight into a process by which affect produces biases in generalized reward expectations.

## Author summary

Positive affective states are associated with optimistic future expectations, and negative affect is associated with pessimistic future expectations. However, the cognitive mechanisms that underpin these affect-congruent shifts in reward expectations remain unclear. To investigate this question, we focused on feature-based attention, the process by which attention to the different features of a stimulus influences the estimated value of that stimulus. We formulated a new compound generalisation paradigm to investigate how individuals allocate attention to high- versus low-value components of novel compound stimuli, and adopted a multi-method approach combining eye-tracking and computational modelling of behavioural data. Crucially, our central experimental manipulation was a controlled between-subjects laboratory affect induction during the generalisation phase of the task. The results of this study clearly identify feature-based attention as a cognitive mechanism by which affective states influence reward expectations: in positive

eye-tracking data developed in-house (available at https://github.com/nivlab/NivLink).

**Funding:** This work was supported by funding from the NIH to YN and salary support from the NHMRC to DB. The funders had no role in study design, data collection and analysis, decision to publish, or preparation of the manuscript.

**Competing interests:** The authors have declared that no competing interests exist.

affective states, participants attended more strongly to high-value cues within compound stimuli (and therefore formed more optimistic reward expectations for the compounds). In negative affective states, the converse was true: participants attended more strongly to low-value cues within compound stimuli, and therefore formed more pessimistic reward expectations for the compounds. These behavioural and modelling findings were separately corroborated by evidence from eye-tracking data.

## Introduction

Affect is deeply intertwined with expectations about future reward and punishment. Whereas psychological wellbeing and positive affect are linked with optimistic future expectations, negative affect is associated with pessimistic expectations [1, 2]. Clinically, various symptoms of mood disorders can be viewed as changes in expectation formation: mania is associated with grandiose positive expectations [3, 4], whereas depression and anxiety are marked by pessimistic expectations about future reward and punishment [4, 5]. It is therefore critical to understand how affect might influence cognitive processes that underlie expectation formation.

One process by which humans and other animals form reward expectations for novel stimuli is feature-based generalization [6–9]. In choosing whether to rent an apartment, for example, a prospective renter might consider features such as its floorspace, the quality of its furnishings, and its neighborhood. Using a weighted combination of these features, they can form a *generalized reward expectation* (i.e., a reward expectation formed via feature-based generalization) that determines their willingness to rent the apartment.

How people form generalized reward expectations for a novel stimulus depends on how they allocate attention to its different features. For example, if a prospective renter were to attend more to positively valenced stimulus features (i.e., features of the apartment that are predictive of reward, such as good-quality furnishings and appliances), they would form a higher reward expectation for the apartment than if they attended more to its negatively valenced features (such as a dangerous neighborhood or small floorspace). *Feature-based attention* [10–13] is therefore a crucial determinant of reward expectation: if attention to a stimulus is biased toward its high-value features, then reward expectations will be inflated as a result, and vice versa if attention is biased towards low-value stimulus features.

Separately, human visual attention is known to be modulated by affect in an *affect-congruent* manner, such that subjects preferentially attend to visual stimuli that are congruent with their affective state. Whereas positive affective states facilitate attentional selection of positively valenced visual stimuli [14–16], negative affect is associated with increased attention to negatively valenced stimuli [16–18]. These results are consistent with affect-congruent information-processing biases across a number of species and cognitive domains (e.g., [19–23]), though it should be noted that other, more complex patterns of interaction between affect and attention have also been demonstrated [24–28].

If affective biases alter value-based attention in the same affect-congruent manner as they do visual selective attention, we would predict affect-congruent changes in generalized reward expectations as a result. That is, we would predict positive affect to produce higher generalized reward expectations (via increased attention to high-value stimulus features), and negative affect to produce lower reward expectations (via increased attention to low-value stimulus features). This interaction between affect and attention should alter choice patterns, a hypothesis that has not previously been tested in the domain of decision-making.

To test this hypothesis, we combined eye-tracking with a novel *compound-generalization* task [29–31]. In a learning phase, participants first learned to associate a set of simple visual cues with differing reward probabilities; then, in a generalization phase, participants were presented with choices involving novel compounds composed of pairs of simple cues from the learning phase. To test the effects of affect on reward expectation, participants experienced either a positive, neutral, or negative affect induction during the generalization phase of the task. We hypothesized that we would observe an increase (decrease) in generalized reward expectations for novel compounds following a positive (negative) affect induction, and that the mechanism for this effect would be affect-congruent changes in attention (i.e., increased attention to high-value cues in the positive affect group, and vice versa in the negative affect group).

## Results

120 adult participants (77 female, 43 male; mean age (SD) = 21.34 (4.40)) completed a novel "space mining" compound-generalization task designed to assess the effects of affect on the distribution of attention over different cues within compound stimuli (see Fig 1 and Methods). Participants were told they would be mining minerals from different "planet" stimuli, where each planet could contain either a valuable mineral (termed "positivium") or a worthless mineral ("negativium"). Different planets were denoted by different cues (rune symbols), each of which was associated with a different probability of yielding the valuable mineral. The experiment used a between-participants design for affect inductions, with participants pseudo-randomly allocated to receive either positive, neutral, or negative affect inductions during the task.

The task was divided into three phases (Fig 1A). The first phase involved learning the values of six different cues, and the second assessed whether participants had successfully learned these values. In the third phase, the critical test of our hypotheses, participants chose between compounds of the previously learned cues with no new outcomes. These generalization choice trials were interspersed with an affect-induction procedure (positive, neutral, or negative, between participants; see Methods). Eye-tracking was conducted throughout all three phases of the task, and subjective current mood was elicited at the start and end of each block. We also collected several self-report measures quantifying individual differences in current mood state and potential symptoms of mood disorders (trait depression and hypomania as well as state positive and negative affect; see Methods for further information), since we reasoned that these individual differences might moderate the effects of an affect induction on behavior [32, 33]).

### Participants accurately learned values of simple cues

In the first phase of the task, participants learned the reward contingencies of six distinct rune cues via a Pavlovian learning procedure (Fig 1B). For each participant, there were two low-value cues (reward probability of 25%; hereafter denoted *L*), two medium-value cues (reward probability of 50%; denoted *M*) and two high-value cues (reward probability of 75%; denoted *H*). In the second phase (Fig 1C), we verified that participants had successfully learned to distinguish between the different cues by offering them choices between all pairs of cues (without feedback, to avoid unequal exposure to the outcomes of each cue type due to the participant's specific choices).

Participants displayed good overall learning (mean proportion correct in simple cue test trials = 0.89, SD = 0.11). As expected given that the learning phase of the task preceded the subsequent between-participants affect induction, there was no evidence for a significant difference in learning between different affect induction groups ($\beta$ = −0.12, $p$ = .41, mixed-effects logistic

## A. Overview of task structure

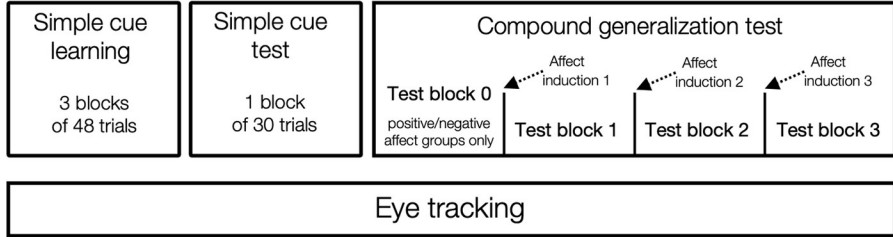

## B. Trial schematic: simple cue learning

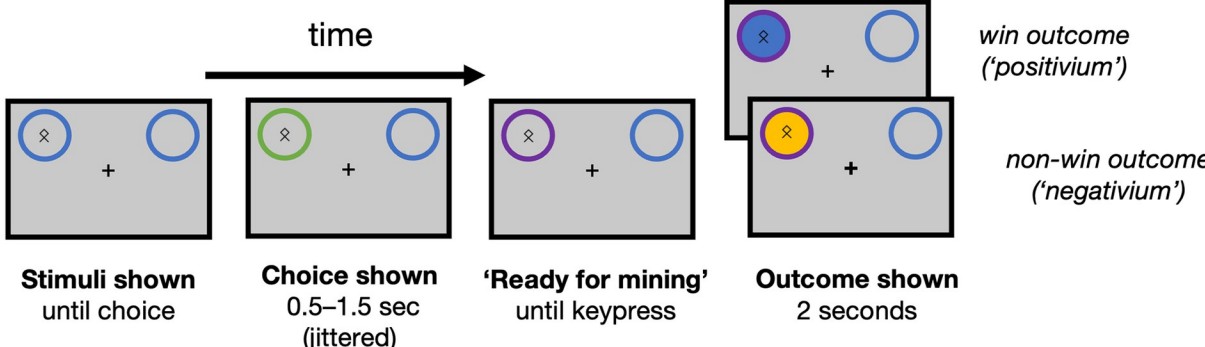

## C. Trial schematic: simple cue test and compound generalization

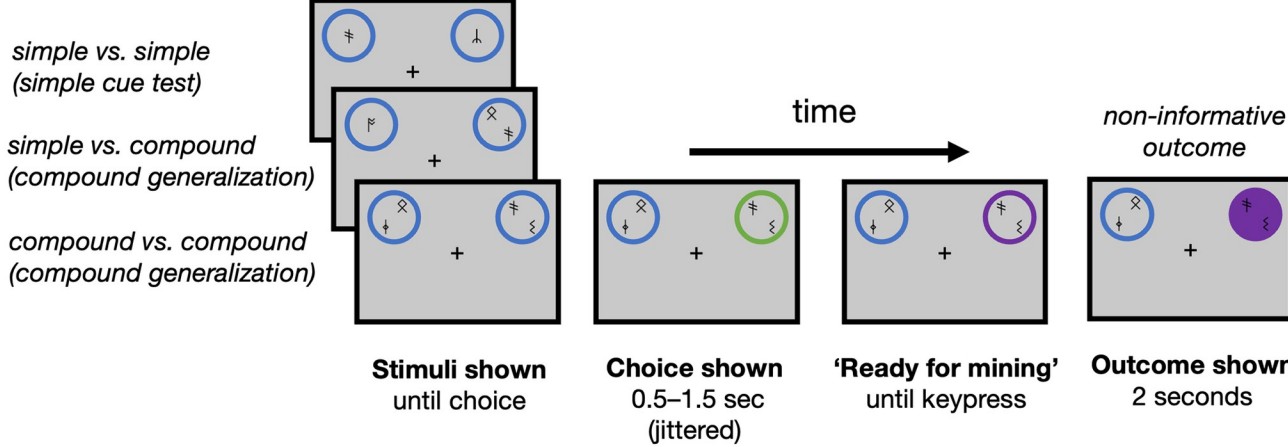

**Fig 1. Task structure. A**. The task comprised three phases, with affect inductions (90-second emotional film clips) presented prior to each of three blocks in the compound-generalization phase. For positive/negative affect groups, the induction was preceded by one additional compound-generalization block to measure generalization in neutral affect. **B**. Sequence of visual events for phase 1 (simple cue learning). One cue was presented on each trial, to the left or right of a central fixation cross; after a "planet" was selected for "mining", its outline color changed to green to indicate the participant's selection. The chosen planet's outline color changed again to purple after a jittered interval, signifying that the planet could now by mined. At this point, the participant could press any key to reveal the outcome, which was indicated both by a change in planet fill color to blue (win) or orange (non-win) and a distinctive win or non-win sound. **C**. Sequence of visual events for task phases 2 and 3 (simple cue test and compound-generalization test). The structure of each trial was the same as in phase 1, except that two stimuli were presented and the participant could choose between them, and feedback was uninformative (a neutral purple color and neutral sound). Three example cue configurations are shown, representing (from top to bottom) a simple-versus-simple choice in the simple cue test phase, a simple-versus-compound choice in the compound-generalization phase, and a compound-versus-compound choice in the compound-generalization phase. Cues in compound stimuli were symmetrically offset from the center of the stimulus, ensuring that the distance of all cues from fixation was approximately equal.

regression; Fig 2A). There were also no significant associations between performance at test and individual differences in either baseline self-reported mood valence (Spearman $\rho = .05$, $p = .56$) or arousal (Spearman $\rho = .05$, $p = .60$).

We also assessed post-choice mining times (the response time for revealing the outcome after the chosen planet became ready to mine). Consistent with a general approach bias towards higher-value stimuli, mining time was significantly modulated by cue value during the simple cue test ($\beta = -0.09$, $p = .03$, mixed-effects linear regression), with a faster mining time for higher-value simple cues (mean response time [SEM] for high-value cues: 505.5 ms [19.3]; medium-value cues: 519.9 ms [21.5]; low-value cues: 548.8 ms [25.3]).

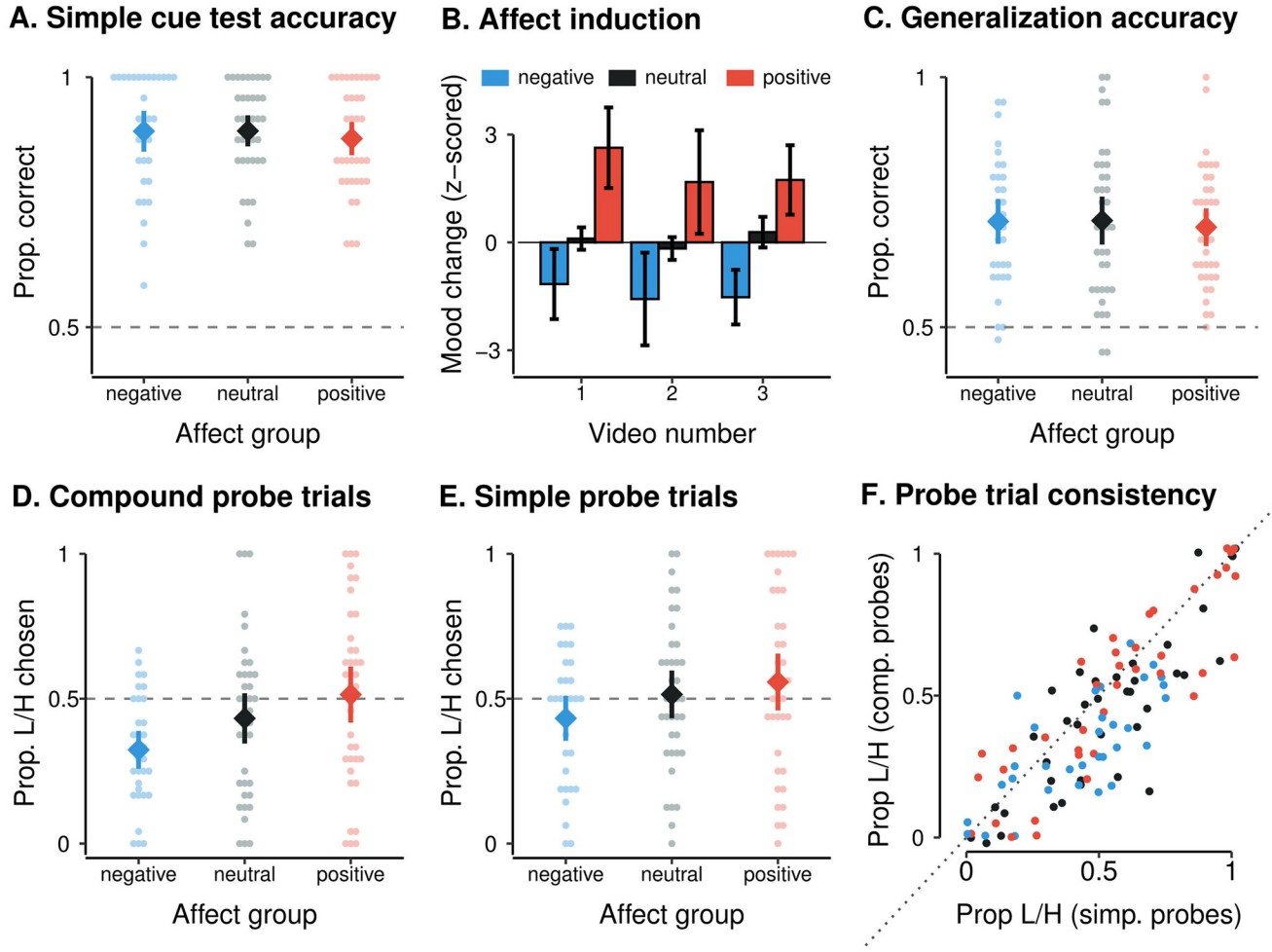

**Fig 2. Behavioral results. A**. The simple cue test evidenced no significant between-groups differences in choice accuracy (i.e., proportion of trials on which the higher-value cue was chosen). **B**. As expected, there was a significant effect of affect-induction videos on self-reported mood valence ($\beta = 2.63$, $p < .001$; mood reports standardized within subject with reference to the average between-block mood change in the learning blocks, where no videos were presented). **C**. In the compound-generalization phase, there were no significant between-groups differences in choice accuracy (the proportion of trials in which the compound with the higher mean value was chosen). **D**. There was, however, a significant between-groups difference in preference for the *L/H* stimulus as opposed to the *M/M* stimulus in compound probe trials ($\beta = 0.79$, $p < .001$). **E**. There was also a significant between-groups difference in preference for the *L/H* stimulus as opposed to the *M* cue in simple probe trials ($\beta = 0.56$, $p = .004$). **F**. Preference for the *L/H* stimulus in simple probe trials and preference for the *L/H* stimulus in compound probe trials were positively correlated (Spearman $\rho = .81$, $p < .001$). Dot color reflects participant condition as per subplots A-E. For all panels, groups are presented as mean ± 95% CI. Points in scatterplots represent condition means for individual participants, and are jittered to prevent overplotting (horizontal jitter in subplots A-D, 2% horizontal and vertical jitter in subplot F).

## Affect inductions successfully induced positive and negative affect

In the third phase of the task, participants chose between pairs of options involving novel compounds of cues from the earlier learning phase. To measure the effects of affect on compound generalization, this phase of the task was conducted with a between-participants video-based affect induction, such that each participant saw either three positive videos, three neutral videos, or three sad videos (one video at the start of each block). Further details of affect-induction stimuli, as well as of a separate affect-induction validation experiment, are presented in the Methods and in S1 Text, Section A. Participants in the positive and negative affect-induction groups also completed an additional block of compound-generalization choice trials prior to the first affect induction. This enabled our computational modelling analyses of behavior (see below) to quantify the effects of the affect induction as a within-subjects change from baseline for these participants (see Methods).

As expected, a mixed-effects linear regression revealed that the affect induction significantly modulated the valence of participants' self-reported mood ($\beta = 2.63$, $p < .001$; Fig 2B), but did not modulate their self-reported arousal ($\beta = 0.44$, $p = .16$). For both valence and arousal, there was no significant effect of video number on self-reported mood change (valence: $\beta = -0.22$, $p = .23$; arousal: $\beta = -0.09$, $p = .39$), and no significant interaction between video group and video number (valence: $\beta = -0.21$, $p = .34$; arousal: $\beta = -0.07$, $p = .60$), suggesting that the influence of videos on self-reported mood did not change over time.

## Participants treated compound stimulus values as weighted averages of simple cue values

After each affect induction, participants were presented with novel compound stimuli comprised of *pairs* of the simple cues that they had learned about previously (Fig 1D; see Methods and S1 Text, Section B for further information on the composition of stimulus choice pairs). Based on previous studies of compound generalization, and given no explicit instruction on how to estimate the values of these new stimuli, we assumed that participants would estimate the value of a compound stimulus as a weighted average of the values of the cues within the compound [6, 9]. A qualitative prediction of a weighted-average rule is that the value of a compound stimulus should be proportional to the mean of the values of its constituent cues (e.g. an *L/H* [25/75] stimulus should be treated as more valuable than an *L/M* [25/50] stimulus, but less valuable than an *M/H* [50/75] stimulus). (This prediction encounters a limit as the weighting on any single cue approaches 100%, in which case the value of the compound would be solely determined by the value of the single attended cue.

Fig 2C shows this predicted qualitative pattern in our data. In choices between compound stimuli with different mean cue values, participants tended to choose the stimulus with the higher mean value (mean choice proportion for higher-mean-value stimuli (SD) = 0.71 (0.13); $p < .001$, Wilcoxon test against chance). This pattern did not differ significantly across affect groups ($\beta = -0.09$, $p = .38$, mixed-effects logistic regression) or across different blocks of the task ($\beta = -0.003$, $p = .95$, mixed-effects logistic regression). In particular, we observed the predicted qualitative pattern for the choices described above: participants showed a significant preference for the higher-mean *L/H* stimulus over the lower-mean *L/M* stimulus when these were the two choice options presented (*L/H* stimulus chosen on 63% of such trials; $p < .001$, one-sample *t*-test against chance), and a significant preference for the *M/H* stimulus over the *L/H* stimulus (*M/H* stimulus chosen on 70.2% of trials; $p < .001$, one-sample *t*-test against chance).

For simple-versus-compound trials, we found that participants' choice behavior was sensitive to the value of both cues within each compound: we found that the *M* simple stimulus was

chosen significantly more frequently in $M$ vs. $L/M$ trials (mean choice proportion = 61.2%) than in $M$ vs. $L/H$ trials (mean choice proportion = 48.3%; $t(116)$ = 5.13, $p < .001$, paired samples $t$-test). By contrast, participants chose the $M$ simple stimulus significantly less frequently in $M$ vs. $M/H$ trials (mean choice proportion = 24.9%) than in $M$ vs. $L/H$ trials ($t(116)$ = −9.30, $p < .001$, paired samples $t$-test). These results indicate that participants did not simply attend to the highest- or lowest-value cue in a compound, but attended to both its constituent cues when estimating the value of a compound stimulus.

Finally, as in the simple-cue test phase, planet mining time for compound stimuli was significantly modulated by mean cue value ($\beta$ = −0.18, $p$ = .04, mixed-effects linear regression), such that participants were significantly faster to mine higher-mean-value compounds (mean response time [SEM] for $M/H$ compounds: 464.4 ms [16.1]; for $L/H$ or $M/M$ compounds: 470.0 ms [14.4]; for $L/M$ compounds: 495.5 ms [17.7]). These findings support the explanation that participants estimated compound cue values using a linear weighted average.

## Affect inductions produced affect-congruent biases in valuation of compound stimuli

The primary goal of the compound-generalization phase was to test participants' allocation of attention to the different simple cues contained within each compound stimulus. The main trials of interest for this analysis were 'probe trials', in which participants chose between an $L/H$ compound stimulus and either an equal-mean simple $M$ stimulus (*simple probe trials*) or an equal-mean compound $M/M$ stimulus (*compound probe trials*). In these trials, the value the participant assigned to the $L/H$ stimulus—and, therefore, their preference for this stimulus over the alternative—depended on the respective attention weights assigned to the $L$ and $H$ cues. If more attention was paid to the $L$ cue, the estimated value of the compound $L/H$ stimulus would be lower, and the participant would therefore tend to chose it less frequently than the alternative stimulus (either $M$ or $M/M$), and vice versa if more attention was given to the $H$ cue.

Crucially, mixed-effects logistic regression analyses revealed a clear modulation of preference for the $L/H$ stimulus by affect group, both in compound probe trials (Fig 2D; $\beta$ = 0.79, $p < .001$) and in simple probe trials (Fig 2E; $\beta$ = 0.56, $p$ = .004). As we hypothesized, this modulation was consistent with affect-congruent shifts in attention such that across trial types, preference for the $L/H$ stimulus was strongest in the positive affect group, intermediate in the neutral affect group, and weakest in the negative affect group.

For compound probe trials, there was a significant interaction between affect group and block number, such that the effect of affect induction on choice behavior weakened over time (S1 Text, Section C); compound probe trials: $\beta$ = −0.27, $p$ = .006; cf. simple probe trials: simple probe trials: $\beta$ = −0.19, $p$ = .07). The main effect of block number on choice behavior was not significant for either trial type (compound probe trials: $\beta$ = 0.14, $p$ = .07; simple probe trials: $\beta$ = −0.08, $p$ = .30). There was no evidence for an association between any individual-difference measure and performance in either probe trial type, and no interaction between any individual difference measure and the strength of the effect of the affect induction on choice behavior (all $p > .10$; see Methods for a list of individual-difference measures).

Choice behavior on both simple and compound probe trials showed good internal consistency (Spearman-Brown-corrected split-half reliability: simple probe trials $\rho$ = .85, compound probe trials $\rho$ = .88). In particular, preference for the $L/H$ stimulus was strongly positively correlated across simple and compound probe trials (Fig 2F; Spearman $\rho$ = .81, $p < .001$). These psychometric results suggest that both probe trial types reliably measure the same underlying behavioral phenotype.

Lastly, we also investigated between-groups differences in overall preferences for simple versus compound stimuli over all choice pairs (S1 Text, Section C). This analysis provides a measure of whether the affect induction influenced preferences for compound stimuli in general (including but not limited to simple probe trials). Similar to the observed effects in simple probe trials alone, we found evidence for an affect-congruent modulation of preference for compound stimuli, such that preference for compound stimuli was strongest in the positive affect group, intermediate in the neutral affect group, and weakest in the negative affect group ($\beta = 0.33$, $p = .01$, mixed-effects logistic regression). Once again, there was a significant interaction between this effect and block number, such that between-groups differences became less pronounced over time ($\beta = -0.15$, $p = .03$, mixed-effects logistic regression). However, note that since all non-compound cues were M cues, simple preference for the H or L cues could manifest as preference for compound stimuli in this analysis. In our modelling of task behavior (below) we tease apart these two effects.

The interactions between affect group and block number for both simple and compound probe trials suggest that the affect induction's effect on behavior weakened over time, despite there being no block-wise differences in the effect of videos on self-reported affect. It is possible that initial surprise at the occurrence of compound stimuli might have resulted in greater overall salience of compounds in early compound-generalization blocks. In this case, the greater effect of the affect induction on behavior in earlier blocks might have resulted from the fact that more attention was initially paid to these compound stimuli in absolute terms [34], resulting in greater downstream effects of the affect induction on behavior. See S1 Text, Section C, for analyses of choice behaviour in non-probe trials during the compound generalization phase of the task.

## Computational modelling of behavior captured the effects of affect on attention

We next turned to computational modelling to dissect the influence of affect on compound generalization. We used a modelling framework (see Methods) in which the values of compound stimuli were assumed to be an attention-weighted linear combination of the values of simple cues within the compound (denoted $v_j$), multiplied by a parameter $\phi$ that captured the non-specific tendency for participants to prefer compound stimuli to simple stimuli or vice versa:

$$V(\text{compound}) = \phi \cdot \sum_{j=1}^{N} A_j \cdot v_j \qquad (1)$$

The attention weights ($A_j$) for each cue in a compound were allowed to vary as a function of the *relative value* of each simple cue, such that attention might be biased either towards or away from high-value cues, with individual differences controlled by an $\alpha_V$ parameter. The relative influence of the simple cue values were then normalized to ensure that the attention weights for all cues within a compound summed to 1:

$$A_j = \frac{\exp\left(\alpha_V \cdot v_j\right)}{\sum_{k=1}^{N} \exp\left(\alpha_V \cdot v_k\right)} \qquad (2)$$

Eq 1 presents the model in its general form. For compounds of two cues like those in the present study, this equation can also be expressed as $V(\text{compound}) = \phi \cdot (A_1 \cdot v_1 + A_2 \cdot v_2)$, where $A_2 = (1 - A_1)$ and $0 \le A_1, A_2 \le 1$.

We assumed that choices between stimuli were made on the basis of stimulus values using a softmax policy-mapping function, with individual differences in the stochasticity of the value-to-policy mapping controlled by the inverse temperature parameter $\beta$. As such, the full framework accounted for individual differences in behavior using three parameters per participant: the compound preference parameter $\phi$, the attention-to-value parameter $\alpha_V$, which was our primary parameter of interest, and the softmax inverse-temperature parameter $\beta$.

We conducted a model-recovery analysis to ensure that our experimental design was able to differentiate between models in which different combinations of these parameters were free to vary (see Table 1). That is, we simulated data from each model, and tested whether each dataset was correctly identified as having been generated by the relevant model using our model-comparison procedure. The results, reported in S1 Text, Section D, indicated that the models were appropriately identifiable on the basis of data collected in our task. S1 Text, Section D also reports the results of a parameter-recovery analysis, which indicated that all of the parameters in each model could be estimated with adequate accuracy by our model-fitting procedure.

**Modelling of neutral-affect data.** First, we sought to determine whether all the above parameters were necessary to account for participants' task behavior *in the absence of an affect induction*. That is, we compared the full model to simpler models that involve subsets of the three parameters using all choices of participants in the neutral affect group and choices of participants in the positive and negative groups in the pre-induction baseline block only.

To do this, we formulated four models in which different subsets of the three parameters were free to vary across participants (Table 1). We first conducted a model-recovery analysis to ensure that our experimental design was able to differentiate between these models. For this, we simulated data from each model, and tested whether each dataset was correctly identified as having been generated by the relevant model using our model-comparison procedure. The results, reported in S1 Text, Section D, indicated that the models were appropriately identifiable on the basis of data collected in our task.

We then fit the four models to choice data using Hamiltonian Monte Carlo estimation (see Methods), and compared models using standard comparison statistics for hierarchical Bayesian models (see Methods). The results, reported in Table 2, indicated that the full model (#4), in which all three parameters were free to vary across participants, provided the best overall account of participants' choices in phases 2 and 3 of the task, even when accounting for (and penalizing in the model comparison) additional degrees of freedom. This suggested that each of the parameters corresponds to a meaningful dimension of individual variability in behavior.

**Modelling the effects of the affect induction.** We next considered which of the two major parameters in Model 4 ($\alpha_V$ and $\phi$) were modulated by the affect induction. We did not consider modulations of the softmax inverse temperature parameter $\beta$, since variation in this

**Table 1. First-stage computational models.**

| Model # | Description | Free parameters |
|---|---|---|
| 1 | Baseline | $\beta$ |
| 2 | Baseline with compound multiplier | $\beta, \phi$ |
| 3 | Value-weighted averaging | $\beta, \alpha_V$ |
| 4 | Value-weighted averaging with compound multiplier | $\beta, \phi, \alpha_V$ |

Note: Simplified models (i.e., those without the full set of free parameters, #1–3) were created by setting non-free parameters to the parameter values necessary to eliminate their effect (0 for the additive parameter $\alpha_V$; 1 for the multiplicative parameter $\phi$).

**Table 2. Goodness of fit of models to neutral-affect data (model numbers as per Table 1).**

| Model number | n free parameters per participant | WAIC | ΔWAIC (Std. Err.) |
|---|---|---|---|
| 1 | 1 | 7559.95 | 1282.80 (57.15) |
| 2 | 2 | 7046.11 | 768.95 (46.21) |
| 3 | 2 | 6319.98 | 42.82 (14.69) |
| 4 | 3 | 6277.15 | 0 (-) |

WAIC: Watanabe-Akaike Information Criterion, presented on a deviance scale such that lower numbers indicate better model fit. ΔWAIC: the difference between the WAIC of each model and that of the best-fitting model (model 4), calculated as per [35] using the `loo` package for R.

parameter can simply be an index of a model's overall goodness of fit. We compared the best-fitting model from the first stage (Model #4) to a set of more complex models in which the affect induction produced a shift in some or all model parameters (see Methods). In addition, we considered the possibility that the influence of the affect induction on model parameters decayed over time, in line with our finding that the strength of affect's influence on behavior weakened over time. Unlike in the first-stage model comparison, in this stage we fit all models to all test-phase choices by all participants. All models were found to be recoverable in a model-recovery analysis (see S1 Text, Section D).

The results of this model comparison (Table 3) supported a model in which affect induction influenced both $\alpha_V$, which controls attention to value, and $\phi$, which controls the preference for compound relative to simple cues. In addition, in the best-fitting model, the strength of affect's influence on these parameters decayed over successive compound-generalization blocks. Specifically, we assumed that this effect decayed according to a power-law decay function with parameter λ. For instance, for $\Delta\alpha_V$, the affect-related change in $\alpha_V$, for instance, the effective parameter value in a given block was computed as $\alpha_V(\text{effective}) = \alpha_V + \Delta\alpha_V \times \lambda^{\text{block}} \times \text{Affect}$, where Affect is a dummy-coded categorical variable denoting the participant's affect-induction condition (with neutral affect as the reference condition; see Eq 7 Methods for further information), such that the effect of affect decayed across blocks. We note that this block-wise change in parameters cannot be attributed to influences of affect on learning, since the compound-generalization phase of the task did not involve feedback, and therefore did not prompt new learning.

Our model fitting assumed that participant-level parameters were drawn from group-level distributions and estimated the means and standard deviations of these distributions from the

**Table 3. Results of second-stage model fit to all compound-generalization data.**

| Model number | Parameters modulated by affect | Blockwise decay in effect | n free parameters per participant | WAIC | ΔWAIC (Std. Err.) |
|---|---|---|---|---|---|
| 4 | None | - | 3 | 13698.13 | 189.33 (27.20) |
| 4a | $\phi$ | No | 4 | 13637.80 | 129.01 (24.99) |
| 4a-λ | $\phi$ | Yes | 5 | 13595.81 | 87.01 (20.70) |
| 4b | $\alpha_V$ | No | 4 | 13598.95 | 90.16 (20.20) |
| 4b-λ | $\alpha_V$ | Yes | 5 | 13552.32 | 43.53 (12.61) |
| 4c | $\phi, \alpha_V$ | No | 5 | 13585.01 | 76.21 (19.81) |
| 4c-λ | $\phi, \alpha_V$ | Yes | 6 | 13508.79 | 0 (-) |

WAIC: Watanabe-Akaike Information Criterion, presented on a deviance scale such that lower numbers indicate better model fit. ΔWAIC: the difference between the WAIC of each model and that of the best-fitting model (model 4c-λ).

data. Fig 3 presents samples from the posterior (i.e., post-fit) distributions for the group means of several of these parameters (means of affect-independent parameters in grey, mean affect-related changes in parameters in purple). The mean of the distribution over the value-attention parameter $\alpha_V$ was negative (though overlapping with zero, see Fig 3A; estimated SD of group distribution = 4.27). Negative values of $\alpha_V$ mean that attention was higher for *lower-valued* cues. The group mean for the affect-related change in this parameter ($\Delta\alpha_V$; Fig 3D) was credibly greater than zero (posterior median = 0.68, estimated probability of 96% that the mean effect size in the population was greater than zero), indicating affect-congruent modulation of attention to value, consistent with the probe-trial results presented in Fig 2D and 2E. That is, as positive affect increased, so did attention to higher valued cues.

The group mean of the $\phi$ parameter was significantly above one (Fig 3B; estimated SD for group distribution = 0.19), indicating an overall preference for compound stimuli rather than those including only one cue. The change in this parameter due to affect, $\Delta\phi$, was numerically negative but not significantly below zero (posterior median = -0.03, estimated probability of

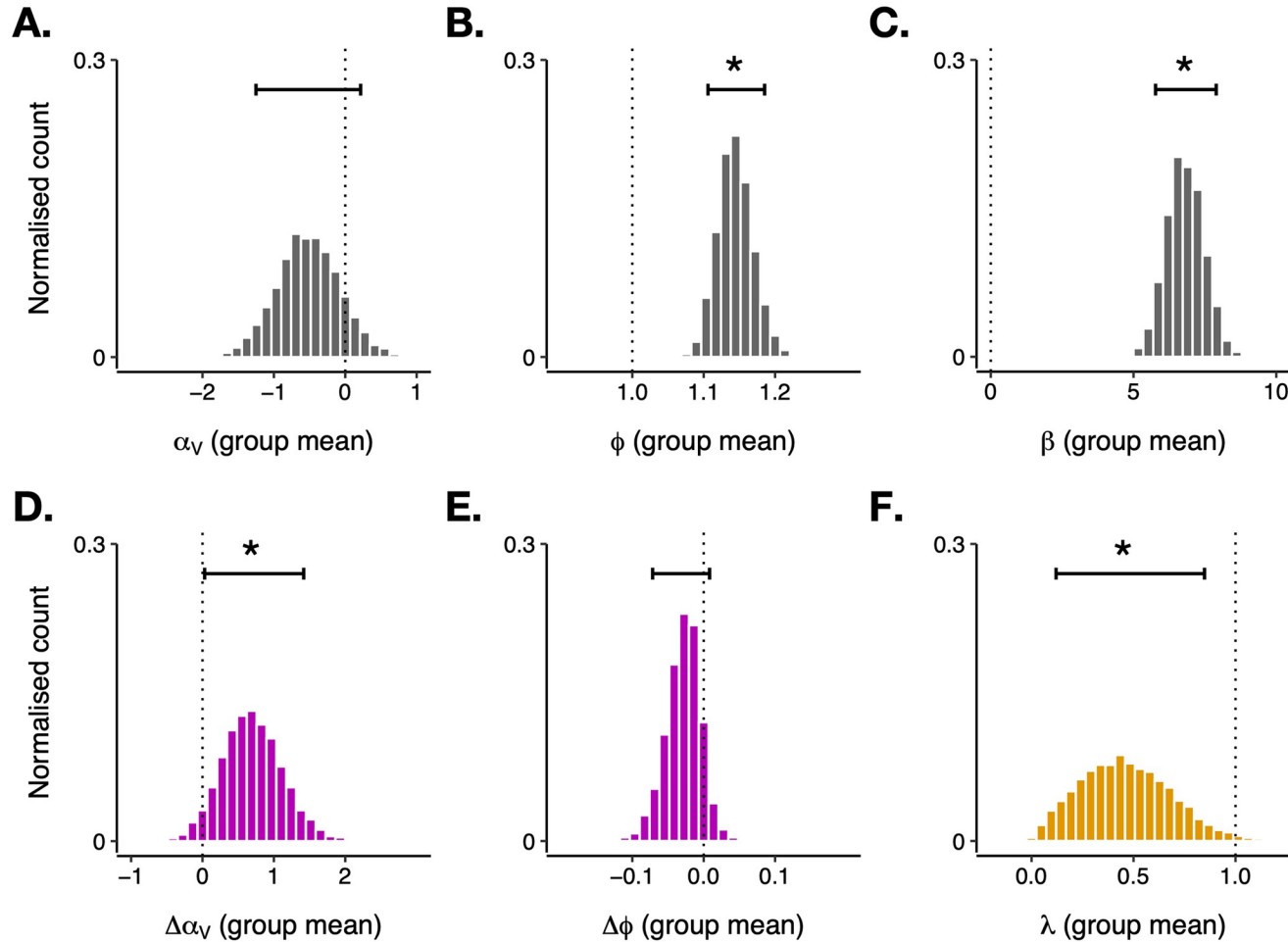

**Fig 3. Estimated posterior distributions for group-mean parameters.** Top histograms: group-level means for the parameters $\alpha_V$ (A), $\phi$ (B), and $\beta$ (C). Bottom histograms: group level means for the effect of affect on $\alpha_V$ (D), the effect of affect on $\phi$ (E), and the block-wise decay in the strength of these effects, $\lambda$ (F). Horizontal error bars denote the 90% credible interval for each parameter, and vertical dotted lines denote the reference value for each parameter (i.e., the parameter value at which there is no influence of the parameter on behavior: 0 for additive parameters such as $\alpha_V$, 1 for parameters that were multiplicative or exponents such as $\phi$ and $\lambda$). Asterisks denote parameters for which there is credible evidence (i.e., estimated probability in excess of 95%) that the group mean is different from the reference value.

89% that the mean effect size in the population was less than zero; Fig 3E). This provides tentative evidence that positive (/negative) affect might have acted to reduce (/increase) participants' overall preference for compound stimuli.

Finally, the posterior distribution of the mean of the decay parameter λ was significantly smaller than one (posterior median = 0.45; Fig 3F), suggesting a considerable block-by-block reduction in the strength of affect's influence on behavior. As a result, the best-fitting model was able to capture the time-dependent effects of the affect induction on behavior (see S1 Text, Section C).

To aid interpretation of these parameter values, Fig 4 illustrates the dynamics of the $\alpha_V$ parameter in the model for different affect groups across time. As mentioned, the negative value of $\alpha_V$ at a group level indicates that in the neutral group, participants tended to attend more to low-value cues rather than high-value cues within a compound. However, this tendency was modulated by the affect induction in an affect-congruent fashion: participants in the negative-affect group tended to attend even more strongly to low-value cues than participants in the neutral group. By contrast, the positive affect induction mitigated the baseline tendency to attend more strongly to low-value cues. The effects of the affect inductions tended to diminish over time such that by the final block of the task, attention to value did not differ strongly across the affect-induction groups.

Our modelling approach, in which affect group was coded as a continuous predictor, implicitly assumed that the effects of positive and negative affect inductions on model parameters were equal in magnitude but opposite in sign. There is no reason why this must be the

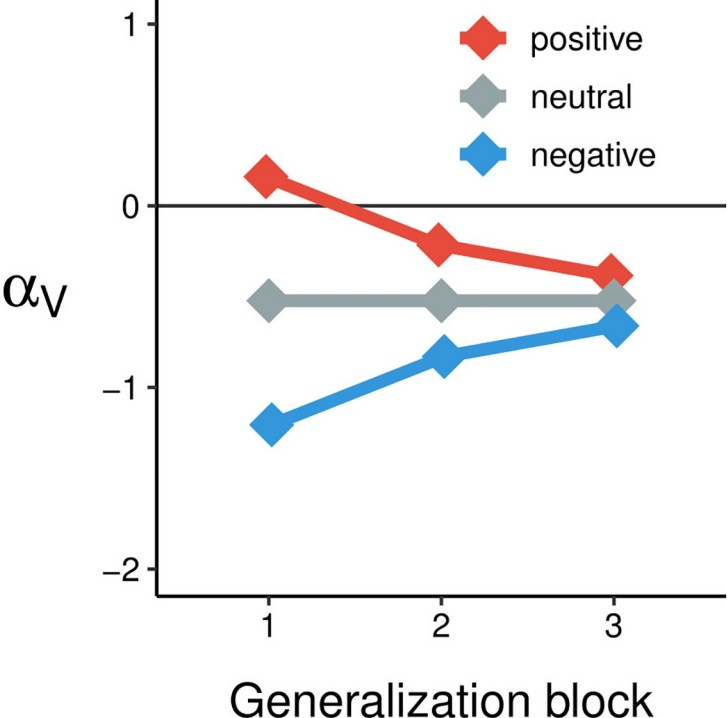

**Fig 4. Illustration of model estimates of attention to valence across time and between groups.** Negative values of $\alpha_V$ indicate more attention to low-value cues; positive values indicate more attention to high-value cues. There was a strong affect-congruent modulation of attention to value in the generalization blocks, the strength of which decayed over time. The dynamics of mood as plotted here were calculated using medians of the posterior distributions of the $\alpha_V$, $\Delta\alpha_V$, and λ parameters.

case. For instance, it is possible that only the positive affect induction influenced a given parameter. We therefore conducted additional model comparisons to test this assumption directly (see S1 Text, Section E). The results of these additional analyses supported the idea that effects of positive and negative affect were symmetric and in opposite directions.

## Eye-tracking data are consistent with affect-congruent modulation of attention

In the analyses above, we interpreted compound-generalization choice behavior in terms of the allocation of attention to different simple cues within compounds. To more clearly determine if differences in choice behavior can specifically be explained by attentional biases, we used eye-tracking data (N = 33, N = 32 and N = 24 participants in the neutral, positive and negative groups, respectively). Specifically, we tested whether affect-congruent modulations of behavior in the compound-generalization task were accompanied by changes in overt attention as measured by patterns of visual fixations.

As an initial validation of our eye-tracking measure, we first tested whether eye-gaze in the simple versus simple trials of the cue test replicated previous findings showing that gaze is biased towards to-be-chosen stimuli [36–38]. Indeed, even before making a choice, participants were more likely to fixate on the to-be-chosen stimulus in simple vs. simple trials (Fig 5A; main effect of subsequent choice status, $F(1, 86) = 19.94, p < .001, \eta_p^2 = 0.19$, ANOVA; no significant interaction between choice and affect group, as expected given that this choice test preceded the affect manipulation). We also tested whether attention in these simple vs. simple trials was biased towards cues in proportion to their value. We found that relative looking time increased as a function of cue value (Fig 5B; main effect of cue value, $F(1, 86) = 31.84, p < .001, \eta_p^2 = 0.27$, ANOVA; no significant interaction). This is consistent with previous demonstrations that attention, as measured by relative looking time, is biased towards higher values [39], and aligned with the fact that participants chose the higher-valued simple cues more often, and fixated more on to-be-chosen stimuli.

To provide mechanistic insight into how affect modulates feature-based attention and its effect on choice, we next turned to analyses of looking time in compound-generalization *probe trials* (that is, trials in which participants chose between the *L/H* compound and either a *M* or *M/M* compound). We specifically asked (1) whether looking time to the high-value cue (*H*) relative to all other cues was predictive of choice, and (2) whether looking time to the high-value cue changed as a result of affect inductions.

We first investigated the overall (group-average) proportion of looking time to low- and high-value component cues during compound-versus-compound trials (in which participants chose between two compound stimuli) and simple-versus-compound trials (in which participants chose between one simple stimulus and one compound stimulus). We found that participants tended to spend slightly more time fixating on the low-value cue (25.8% of total looking time) than the high-value cue (23.9% of total looking time; $p < .05$, paired-sample $t$-test), with the remainder of the looking time spent on the 50/50 stimulus. These results were therefore conceptually consistent with the negative group-level mean for the $\alpha_V$ parameter in computational modelling analyses (which predicts more attention to low-value cues within the *L/H* compound). In simple-versus-compound trials there was no significant overall difference in the proportion of time looking at low- versus high-value cues (40.5% versus 40.0%; $p = .67$; the remainder of the looking time was spent on the 50% simple stimulus).

Then we asked whether the general tendency to look at the high-value cue within an *L/H* compound predicts changes in choice. We used a mixed-effects analysis to test whether, in all probe trials, time looking at the high-value cue relative to all other cues predicted choice of the

                    

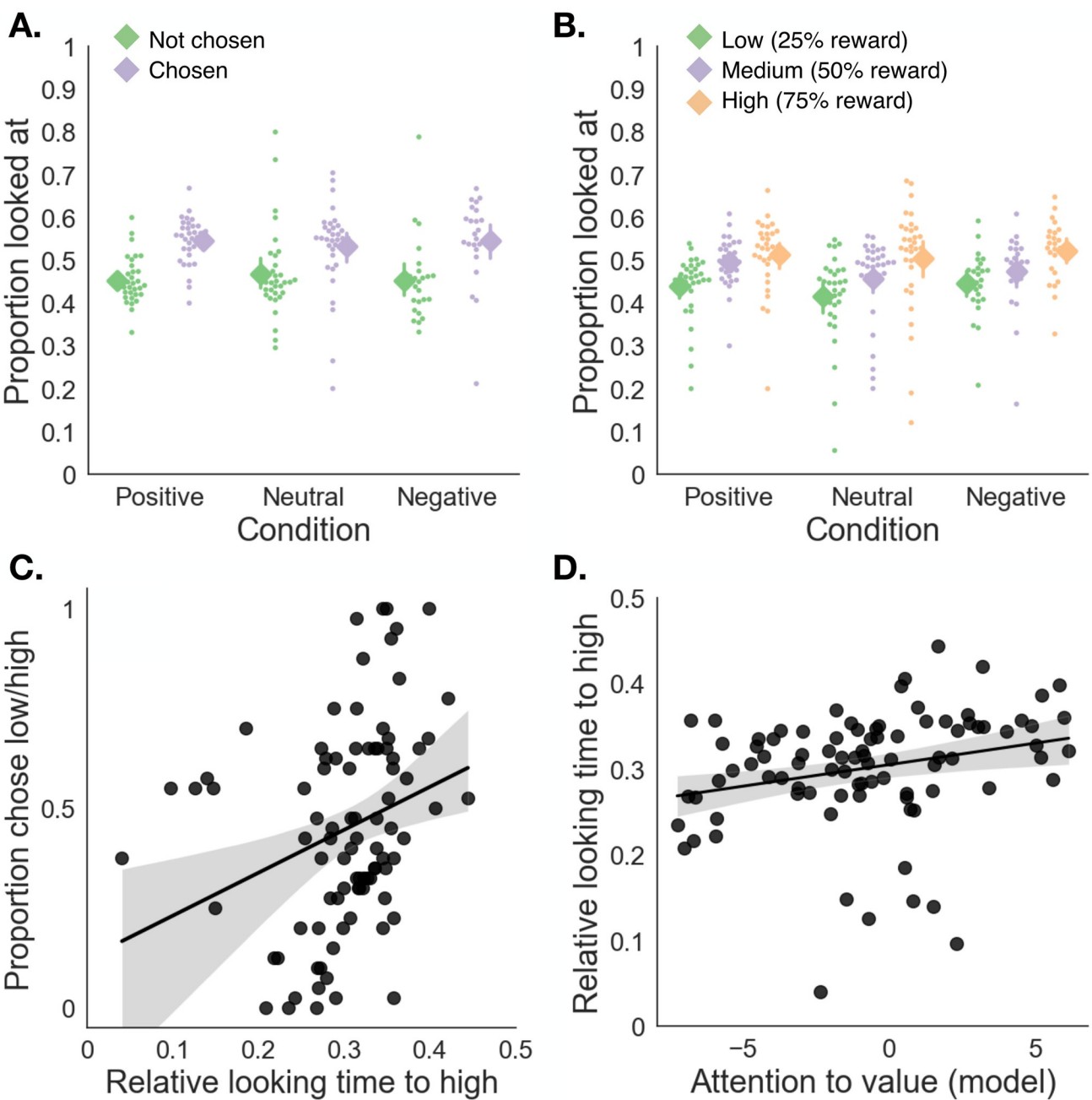

**Fig 5. Eye-tracking validation results. A**: Proportion of time spent looking at each cue in simple versus simple trials, as a function of whether a cue was subsequently chosen or not. For all groups, participants tended to look more at the to-be-chosen cue. **B**: Proportion of time spent looking at each cue in simple versus simple trials, as a function of cue value (25, 50 or 75% probability of reward). For all groups, participants looked longer at more valuable cues. Diamond markers denote the mean of each group and its 95% confidence interval; dots indicate individual participants. **C**: Relative looking time to the high-valued cue within a $L/H$ compound (normalized to the total time looking at all cues in the trial), plotted against the tendency to choose the $L/H$ compound in probe trials. **D**: Attention to value estimated from the computational model, plotted against the relative looking time to the $H$ cue within a $L/H$ compound during probe trials. Data points are color-coded by condition (neutral, positive and negative affect). Each point corresponds to one participant in each of the three groups.

                                    

*L/H* compound, controlling for main effects of affect condition. Consistent with our hypothesis, we found that the more time people spent looking at the high-value cue, the more likely they were to choose the *L/H* stimulus over an *M* or *M/M* stimulus of the same expected value (Fig 5C, $\chi^2(1) = 83.1$, $p < .0001$; likelihood-ratio test for mixed-effects regression; interaction was not significant: $\chi^2(1) = 3.7$, $p = .16$). This finding also held when we instead calculated relative looking time to the *H* cue as a proportion of total time looking at the *L/H* compound (S1 Text, Section F, $\chi^2(1) = 7.54$, $p < .05$; interaction was not significant: $\chi^2(1) = 3.18$, $p = .21$), suggesting that the effect on choice is specifically due to an increase in relative attention to the high over the low-value cue.

To establish the correspondence between our two measures of attention to value ($\alpha_V$, estimated from choice, and relative looking time), we ran a mixed-effects analysis with the model-based attention parameter ($\alpha_V$) as a predictor of relative looking time to the *H* cue relative to all other cues. We found that the model-based attention to value (individually estimated from choice during all compound generalization trials) was a significant predictor of people's tendency to look at the *H* cue during all probe trials (Fig 5D, $\chi^2(1) = 4.69$, $p < .05$; interaction was not significant: $\chi^2(1) = 0.75$, $p = .69$). This result validates our choice of the model-based attention parameter as a metric of value-based attention. The result was not significant when we calculated relative looking time to the *H* cue within the *L/H* compound ($\chi^2(1) = 0.71$, $p = .41$), a point we return to in the discussion.

Finally, we directly investigated the role of positive and negative affect in driving attention by assessing how relative looking time to affect-congruent cues changed following the affect induction. In this analysis, we simultaneously quantified the extent to which the positive affect induction increased attention for the *H* cue, and the negative affect induction increased attention to the *L* cue during probe trials. A mixed-effects model predicting looking time to affect-congruent cues within the *L/H* compound (relative to time looking at all other cues in the trial) showed a significant interaction between probe trial type (simple vs. compound) and affect induction (pre- vs. post-induction; $\chi^2(1) = 5.15$, $p = .02$, likelihood-ratio test for mixed-effects regression). Post-hoc tests revealed that this interaction was driven by a significant increase in time looking at affect-congruent cues in compound probe trials ($\beta = 0.071$, $p = .03$; Fig 6A) but not in simple probe trials ($\beta = -0.073$, $p = .19$; Fig 6B). That is, when participants made a choice between an *L/H* and an *M/M* compound, we found evidence that participants spent more time looking at high-value cues following a positive affect induction, and more time looking at low-value cues following a negative affect induction. This finding aligns with results of the computational modelling analyses, in which we found that the affect induction specifically altered the $\alpha_V$ parameter of the model, which modulates attention to high- versus low-value cues. Given that relative looking time predicts choice (Fig 5C), these results support the overall hypothesis that affect-congruent attention mediates differences in choice between affect conditions. We note that this pattern of results was only evident when calculating looking time as a percentage of gaze to all cues present on the screen; when we instead calculated looking time to affect-congruent cues as a proportion of time looking at the *L/H* stimulus, we did not find a significant effect of the affect induction on looking time (see S1 Text, Section F, and Discussion).

There was one unexpected discrepancy between the computational modelling results and the eye-tracking data. Computational modelling results implied that the modulation of attention by affect occurred equally in both simple and compound probe trials; by contrast, we found evidence for modulation of attention by affect only within compound probe trials in the eye-tracking analyses.

Lastly, we also found evidence for a significant three-way interaction ($\chi^2(1) = 3.95$, $p = 0.047$, likelihood-ratio test for mixed-effects regression; see insets in Fig 6) between probe

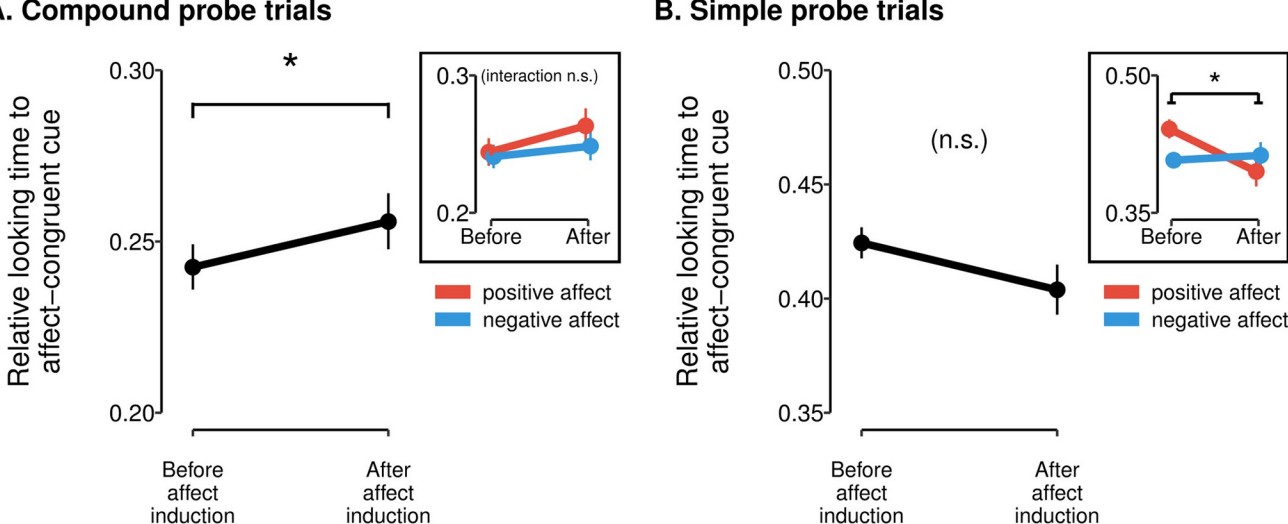

**Fig 6. Affective modulation of relative looking time.** Average relative looking time to the affect-congruent cue (i.e., the low-value 25% cue for participants receiving a negative affect induction and the high-value 75% cue for participants receiving a positive affect induction) pre- versus post-affect induction. Data are plotted separately for compound probe trials (**A**) and simple probe trials (**B**). There was a significant increase in time looking at affect-congruent cues after the affect induction in compound probe trials (main effect of time; $p < .05$, denoted by *), but not in simple probe trials. Insets: data separated by affect-induction condition (red: positive affect; blue: negative affect). The affect by timepoint interaction was significant for simple probe trials ($p < .05$) but not for compound probe trials, though this interaction was driven by group differences in eye-gaze before the affect induction, and as such is of no interest. Error bars denote the standard error of the mean computed based on estimated marginal means from the linear mixed-effects analysis.

type, timepoint (before versus after the affect induction) and affect-induction valence (positive versus negative). Post-hoc tests revealed that this interaction was largely driven by a difference in time looking at affect-congruent cues in simple probe trials *prior to* the affect induction ($\beta = -0.07$, $p = .01$). Since this pre-induction difference is an effect of no interest and did not influence the primary effects of interest within the compound probe trials as described above, we do not interpret this finding further.

## Discussion

### Summary and significance

In this study, we tested the influence of positive and negative affect on participants' behavior in a novel compound-generalization task. We hypothesized that affect would alter reward expectations for unfamiliar compound stimuli via affect-congruent modulation of attention to features of the compounds. As predicted, we found that participants receiving a positive affect induction showed increased reward expectations for compound stimuli relative to those receiving a neutral affect induction, and vice versa for participants receiving a negative affect induction. Computational modelling suggested that this effect was driven by affect-congruent changes in the allocation of participants' attention to high-value versus low-value cues within compound stimuli. These results were corroborated by analyses of participants' gaze-fixation patterns, which revealed an analogous pattern of affect-congruent modulation of overt attention to high- versus low-value features.

Taken together, these results suggest a potential computational and perceptual-level mechanism for the well-documented finding that positive affect is associated with optimistic future expectations (i.e., value), whereas negative affect is linked with pessimistic future expectations [1, 2, 40, 41]. Concretely, systematic affect-congruent biases in attention to high- versus low-

value features of abstract future prospects (e.g., attending to positive reviews of a fine-dining restaurant in a positive mood, versus attention to the large expected expense in a negative mood) would produce increased reward expectancies in positive affective states and lowered reward expectancies in negative affective states. This is consistent with previous studies showing an increased perceptual attention bias towards positive stimuli in individuals that score high on trait optimism, happiness, and proneness to hypomania [33, 42, 43]. By extension, our results may also suggest a potential mechanism for affect-congruent changes in reward and punishment expectations in mania, depression, and anxiety [32, 44–46].

## Converging evidence from choice and eye-tracking data

Our study operationalized attention in two different ways. On the one hand, we operationalized attention as a parameter ($\alpha_V$) within the computational model of behaviour that controlled the degree to which particular cues within a stimulus contributed to the overall value of that stimulus as a function of their value. This was equivalent to the common 'attention weight' formalization of attention in models of choice (see, e.g., [47]). On the other hand, in our eye-tracking analysis we operationalized attention as the proportion of total looking time that was spent looking at each cue during the choice phase of this task [13, 36–38, 48]. We found that there were some discrepancies between model-based measures of attention and eye-tracking measures of attention.

Most notably, our gaze analyses produced an unexpected pattern of results when trials were subdivided into simple probe trials (involving choices between one single-cue stimulus and one compound-cue stimulus) versus compound probe trials (involving choices between two distinct compound stimuli). Consistent with the behavioral and computational modelling results, relative time looking at affect-congruent cues (i.e., looking time for high-value cues after a positive affect induction and for low-value cues after a negative affect induction) increased as a result of the affect induction. This effect was seen in compound probe trials, but surprisingly was not present in simple probe trials. One potential explanation for this unexpected distinction is that compound stimuli pose a more stringent requirement for overt attention (i.e., shift of eye gaze) due to visual crowding [49], whereas simple stimuli are more readily processed using covert attention only (see, e.g., [50, 51]). As a result, our eye-tracking measure may have provided a better estimate of attention in trials where both stimuli were compounds.

We also found that the expected pattern of affect-congruent attention in probe trials depended on whether the looking time measure of attention was computed relative to all cues on the screen, or only captured relative attention between the low and high-value cues within a compound. In our task, this discrepancy suggests a more complex interaction between affect and attention than a simple value-congruence account. For instance, it could be that negative affect increases attention to uncertain cues, thereby indirectly shifting attention away from the high-value cues [52]. Future work might seek to address this alternative explanation by using cues for which values are pre-established (and elicited through self-report or other methods) and not acquired through conditioning.

## Configural and elemental theories of generalization

Independent of the affect induction, we found that in our setting, participants tended to evaluate compound stimuli as linear combinations of the values of their constituent cues. Although weighted cue combination is a common assumption across models of generalization (e.g., [53, 54]), we note that participants were not instructed how to interpret compound stimuli, and never experienced outcomes associated with these stimuli. As such, they could conceivably

have chosen other strategies, such as evaluating the compound using a non-linear XOR cue combination rule (e.g., a compound with either cue A or cue B has high value), as is observed in the biconditional discrimination paradigm [55].

Within the set of possible weighted cue combination rules, we found that participants' behavior during compound generalization was more consistent with cue averaging than with cue-summation. This is notable because cue averaging is predicted by configural models of compound generalization (e.g., [54]), which assume that compounds are treated as unique configurations that accrue value according to their similarity to the values of other stimuli (the simple cues, in this case). By contrast, elemental models of compound generalization assume that expectations for compound stimuli are computed by summing the values of independent features [29, 53, 56, 57]. Here, elemental models would predict that participants would never prefer simple stimuli over compound stimuli (since the summed value of compound cues was always equal to or greater than the value of simple cues in our task). More broadly, whether behavior is consistent with elemental or configural generalization has been shown to depend on experimental context (for review see [31, 58]). In our case, the binary reward distribution that participants were exposed to in the training phase likely favored configural (linear combination) generalization over elemental summation.

Our model predicts an influence of affect on generalized reward expectations for both configural and elemental generalization. Further research is needed to test our hypothesis in the elemental case, for instance, by testing the effects of affect on generalization for compounds comprising cues from different sensory modalities, or comprising cues associated with different outcome amounts rather than different outcome probabilities. In addition, further research is required to investigate the role aversion to uncertainty might play in compound generalization task. In our task, there were at least three different forms of uncertainty that could have affected choices: first, at the cue level, there was uncertainty as to probability of reward for each cue (often called ambiguity), as well as the actual receipt of reward given the probabilistic nature of the task ("risk", which was greatest for the 50% cue). Additionally, for compound stimuli preferences may have been sensitive to uncertainty regarding how to estimate the value of a compound, since participants received no instruction in how to interpret these stimuli, and no feedback in the compound generalization trials. Future work could consider expanding the set of cue-reward contingencies tested in both simple and compound cues to shed further light on this question.

## Related work on affective biases

In our model, the influence of each cue within a compound on reward expectations was dynamically weighted by attention [13]. Attention provides a mechanism for changing the degree of generalization from each simple cue to the compound, and our results suggest that affect-congruent attention may play an important role in how reward expectations generalize across features. However, affect-congruence is not the only principle that might underlie affective modulation of attention. One influential alternative model suggests that positive affect broadens the scope of attention [59]. Affect-related broadening might lead to generalization of reward expectations across a broader set of features, similar to the kind of temporal generalization predicted by several recent normative models of affect [60, 61]. Our results for simple probe trials are consistent with this hypothesis, given that we observed a more equal allocation of attention across cues in a compound in the positive-affect condition relative to the negative-affect condition. More broadly, reward generalization is a promising transdiagnostic computational construct that warrants further investigation as an explanatory computational model of affective disorders [62, 63].

Our findings are also in line with a separate body of research studying affective biases in humans and other animals [23, 64–66]. In these studies, participants are trained with an excitatory and an inhibitory conditioned stimulus that differ from one another along one perceptual dimension (e.g., high-frequency vs. low-frequency audio tones). Affective biases are measured in terms of behavioral responses to subsequent generalization stimuli intermediate between the two training stimuli; across species, positive affect has been associated with an increased positivity bias (i.e., increased generalization of the excitatory conditioned stimulus to intermediate stimuli), and vice versa for negative affect [23, 64–66]. Our results suggest a potential interpretation of these findings in terms of internal attention to appetitive versus aversive internal representations of stimuli (see [67]). Indeed, by reconceptualising interpretation biases as arising from the allocation of attention between latent dimensions of reward-predictive stimuli, affect-congruent attention biases may provide a unifying explanation for effects of affect on judgement and interpretation more broadly (e.g., [68–70]).

## Implications for computational models of affect

The modulation of attention by affect is a phenomenon that presents a challenge to many formal computational models of affect. Ransom et al. [71] recently showed that this is particularly the case for predictive-processing models of affect (e.g., [72]), which conceptualise attention as being driven by the expected precision of sensory inputs, rather than by affective factors. Similarly, although less challenging to core tenets of the theory, affective modulation of attention is also not predicted by recent reinforcement-learning models of affect [60, 73].

Affective modulation of attention might also provide a useful framework for conceptualising the role of anxiety in value-based decision making, which has been shown to drive a bias to learn about punishment-avoidance features when the goal is to avoid threat rather than accrue reward [74]. Our findings demonstrate one manner in which reinforcement-learning models might be extended to account for affective modulation of attention in real-world settings where stimuli and outcomes have multiple features that must be attended to. Further research is required to integrate these ideas into normative accounts of affect, though some preliminary work suggests the form that such a theory might take [62, 75–77].

Finally, our results point to the influence of mood-congruent attentional biases when forming generalized reward expectations. How might such an attentional mechanism be implemented? Attention and memory-based processes both play a role in generalization [78]These accounts are not incompatible: because generalization by definition requires switching between (internal) attention to memory and (external) attention to the sensory environment, one way mood-congruent attentional biases could arise is by directing attention to memories as a function of their valence [79, 80]. This suggests it might be beneficial to integrate affective biases into sampling-based models of value-based and attribute-based attention [81–83]. Sampling-based models propose that attentional effects during value computation, and people's overt fixation patterns, reflect evidence accumulation over internal beliefs about the probability of reward [36–38, 48]. The results of our study suggest that fixation patterns during decision-making are sensitive to affect. Formalising how affective biases arise from information sampling may help integrate empirical findings suggesting that both attention and memory are biased by affect, and in turn mediate affect's downstream effects on choice [19, 43, 84, 85].

To conclude, we provide direct evidence from behavioral and eye-tracking data that experimentally induced changes in affect bias attention towards features predictive of affect-congruent reward outcomes. That is, positive affect leads to increased attention to high-value stimulus features, whereas negative mood produces increased attention to low-value features prospects. These results suggest a potential cognitive mechanism for affect-congruent

modulation of future reward expectations in the general population, as well as for extreme forms of affect-congruent changes in reward expectations associated with psychiatric syndromes including mania, hypomania, and depression.

## Materials and methods

### Ethics statement

This study received ethical approval from the Princeton University Institutional Review Board, and all participants provided written informed consent.

### Participants

We recruited a total of 120 participants (77 female, 43 male; mean age [SD] = 21.34 [4.40], age range 18 to 55), via online advertisements from the general population in Princeton, New Jersey, USA. This study was approved by the Institutional Review Board of Princeton University, and all participants provided written informed consent and had normal or corrected-to-normal vision. Total study duration was approximately 90 minutes per participant. Participants received monetary compensation for their time and travel expenses, plus an incentive-compatible bonus for task performance (mean payment = USD \$19.01, SD = 2.92).

### Design

Participants were allocated to one of three affect groups (positive, neutral, or negative, 40 participants in each group; see *Affect induction procedure* below) in a randomized between-subjects design. This sample size was designed to give in excess of 80% power to detect a moderate effect size of $f^2 = 0.3$ (under standard assumptions of the general linear model and with $\alpha = .05$). Participants completed a behavioral task with concurrent eye-tracking recording, as well as self-report measures of trait depression and hypomania (the General Behavior Inventory (GBI); [86]) and current positive and negative affect (the Positive and Negative Affect Schedule (PANAS); [87]). These individual-difference measures were included because they were identified as potential moderators of the effects of the affect induction on behavior. To ensure that self-report measures were not influenced by the affect induction that occurred during the behavioral task, self-report measures were presented before the task, with the order of the two self-report instruments counterbalanced across participants.

### Behavioral task

The cover story for the behavioral task was a "space mining game" (see Fig 1). Participants were instructed that they would mine minerals from different "planets", and that each planet stimulus could be made of a valuable mineral or a worthless mineral (differentiated by color, blue vs. yellow). Different planet stimuli were marked with different cues (rune symbols), and participants were informed that each cue provided information about the probability that a given planet would yield the valuable mineral when mined. To incentivize learning, participants received a monetary bonus proportional to the amount of the valuable mineral that they mined across the task.

   The task comprised three phases: an initial cue-learning phase, a learning test involving choices between different cues, and a compound-generalization phase involving planet stimuli marked with multiple cues ('compound stimuli'). On each trial, two planet stimuli were presented: one above and to the left of a central fixation cross, and the other above and to the right, and participants chose one of the two planets to "mine" using the computer keyboard (left/right arrow keys). After a short delay (0.5 to 1.5 seconds, uniformly jittered), the chosen

planet became "available to mine" (represented visually by a change in the its border color) and participants could press any key to extract the mineral on the planet. Each trial therefore yielded 3 measures: stimulus choice, choice response time, and reaction time for mining the planet (i.e., time elapsed between when the planet become ready to mine and when the participant mined it). Mining reaction time was taken as an index of approach motivation for the chosen stimulus.

Stimuli were presented in MATLAB with PsychToolbox [88], using a CRT monitor at a resolution of 1024 × 768 pixels. Participants were seated comfortably using a chin rest at a distance of 57 cm from the screen, such that 1 cm on the screen subtended a visual angle of 1 degree. The radius of each planet stimulus was 2.5 centimetres, and each was diagonally offset 13 centimetres from a central fixation cross. Audio was presented via headphones.

**Learning phases.**   In the first phase of the task, participants learned to associate six cues with different reward probabilities (24 exposures per cue, randomly intermixed across 3 blocks of 48 trials each). Two cues were associated with a low probability of reward (25%), two with a medium reward probability (50%), and two with a high reward probability (75%). The allocation of cue images to reward probabilities was randomized across participants. On each trial, a single cue was presented in the center of either the left planet stimulus or the right planet stimulus. Participants were informed that planets without a cue ("empty planets") had zero probability of yielding the valuable mineral, and that it would not benefit them to choose these planets. Consistent with this instruction, participants very infrequently chose planets without a cue (the mean number of empty-planet choices across participants was 2.7 out of a total of 144 first-phase trials). One participant from the negative-affect group who chose the empty planet on more than 25% of learning trials was excluded from further analysis.

The second phase of the task assessed the learning of cue-reward contingencies. Participants completed a single block of choices between the six cues whose reward probability was learned in phase 1 (30 trials total; two repetitions of each possible cue pair). This allowed us to verify that participants had indeed learned to discriminate between cues. To ensure that no further learning took place in this task phase, no informative feedback regarding the outcome of participants' choices was provided (availability of the concealed outcome was denoted by a non-informative purple colour, and participants were still required to "mine" the planet to receive the unknown outcome). Participants were instructed that their selections would still count towards the monetary bonus that they received at the end of the experiment, but that they would not receive trial-by-trial information on the outcomes of their choices (see Fig 1C). Participants still had to mine the chosen planet in this version of the task. Seven participants (five from the negative-affect group and one each from the positive-affect and neutral-affect groups) who did not display above-chance performance levels in this phase of the task (as determined by a binomial test against chance; $\alpha$ = .05, one-tailed) were excluded from further analysis.

**Compound-generalization phase.**   The third phase of the task assessed compound generalization across three blocks of 32 trials each. This phase introduced compound stimuli containing two rune cues rather than one (see Fig 1C). Compound stimuli were introduced into the task without any additional instructions to participants on how they ought to be interpreted. As in the simple-cue test phase, no feedback was provided on the outcome of participants' choices, and participants were informed that their choices would continue to count towards their monetary bonus (though they were not given any information on how the probability of positivium for the compound stimuli would be assessed). By withholding informative feedback on participants' choices, we ensured that we were assessing participants' generalization from previous learning, distinct from new learning of the value of the compounds themselves. This presents a contrast with previous studies of attention to compound stimuli (see

[47, 89] for review), which studied the distribution of attention at learning rather than at choice.

We focused our analysis on two kinds of choice trials that were expected to be especially diagnostic of the distribution of attention within compounds. In *simple probe trials* (16 of 96 trials in the compound-generalization phase) participants chose between a medium-valued simple stimulus (i.e., a planet marked with a single 50% reward cue) and a low/high compound stimulus (a planet marked with one 25% reward cue and one 75% reward cue). In *compound probe trials* (24 trials) participants chose between a medium/medium compound and a low/high compound.

These 40 probe trials provide a critical test of the distribution of attention over cues in a compound. If participants divide attention equally among the cues in a compound stimulus, then a low/high compound stimulus has the same expected value as both a medium-value simple stimulus and a medium/medium compound. If participants allocate more attention to the high-value cue in the low/high compound, however, then they will prefer the low/high compound in both simple and compound probe trials, and vice versa if they allocate more attention to the low-value cue in the low/high compound. Choice behavior on probe trials therefore provides a measure of the distribution of attention to individual cues in the process of compound generalization.

The remaining 56 non-probe trials in this phase of the task were a mixture of different simple-versus-compound trials and compound-versus-compound trials where the two options did not have similar expected values (see S1 Text, Section B for details). This mixture of trial types was designed to yield a rich behavioral dataset that would allow us to fit computational models of the distinct psychological processes at play in compound generalization.

## Affect-induction procedure

We used a video-based procedure to induce either positive, neutral, or negative affect during the compound-generalization phase of the task. Each of the three compound-generalization blocks was preceded by a 90-second video with either positive, neutral, or negative emotional content such that each participant viewed three videos with the same emotional valence. Self-report ratings of affective arousal and valence were collected using affective sliders [90] before and after seeing a video.

We selected a set of 9 videos (3 happy, 3 neutral, 3 sad) from several sources, including affect-induction videos used in previous research [91–93], as well as publicly available clips chosen especially for this study. We confirmed the utility of these videos in inducing the desired affective states in a separate validation study conducted on Amazon Mechanical Turk. Full details of the specific videos used, as well as the results of the validation study are provided in S1 Text (section A).

To measure baseline compound-generalization behavior in the positive-affect and negative-affect groups (i.e., behavior prior to the affect induction), participants in these two groups completed an additional compound-generalization block prior to the affect induction. This additional baseline block comprised 32 trials (16 simple-probe trials and 16 compound-probe trials). Participants in the neutral-affect group did not complete this baseline block, since we assumed that these participants' performance was unaffected by the (neutral) videos that they watched. Indeed, the neutral group showed a flat mood profile throughout the compound-generalization phase (Fig 2B), and as such, we were able to treat task performance for participants in this group as being at 'baseline' affect levels throughout the task.

Since no affect manipulation was conducted during the learning phase of the task, all participants were at a baseline mood level during the acquisition of cue-reward contingencies. This

ensured that between-groups differences in choice behavior during the compound-generalization phase of the task were attributable to the effects of the affect induction on compound generalization itself, rather than any effects of affect on learning. This is important given a large body of work that has shown complex effects of affect on learning across a number of tasks (e.g., [94, 95]).

## Eye-tracking

**Data collection.** Eye-tracking data were acquired using an infrared eye-tracker (SR Research EyeLink 1000 Plus), at a sampling rate of 500 Hz. Fixation points were calibrated prior to the task, and to control within-experiment drift we used automatic drift-correction after each block to re-calibrate fixations to the center of the screen. Before stimuli appeared on each choice trial, participants were required to maintain fixation within 200 pixels of the center of the screen for 1 second (i.e., stimuli did not appear on the screen until this condition was met). This ensured that participants began every choice trial fixating at a location that was approximately equidistant from the two stimuli. Raw output files were analysed using NivLink, an open-source Python package for preprocessing EyeLink eye-tracking data developed in-house (available at https://github.com/nivlab/NivLink).

**AoI selection and quality control.** Our preprocessing and quality-control protocols for eye-tracking data are detailed in S1 Text, Sections G.

Each simple cue was associated with an elliptical or semi-elliptical area of interest (AoI) defined around the center of its corresponding planet stimulus (S1 Text, Section H). The major axes of the ellipses were angled at 45 and -45 degrees from the vertical for the left and right stimulus respectively. To account for the different visual properties of simple and compound stimuli, we used a larger ellipsis around the compound stimulus (exactly twice the area of the simple-cue ellipsis), and created cue AoIs by dividing the ellipsis in two along the minor axis (S1 Text, Section H). This procedure ensured that while each cue could be enclosed either in a full or half-ellipse, the total area of its associated AoI was held constant across stimulus configurations.

In principle, the center of each ellipsis should coincide with the center of each planet stimulus. However, eye-tracking studies suffer from significant post-calibration drift, which typically increases with the length of the experiment [96]. To mitigate this issue, the center of each ellipsis was manually determined blockwise in post-processing by two independent raters (S1 Text, Section G).

As a first quality control step, we excluded from subsequent eye-tracking data analyses those participants for whom inter-rater agreement was low. Specifically, we excluded subjects for whom mean inter-rater disagreement was more than 1.5 of the inter-quartile range (IQR) above the third quartile of overall disagreement (in practice, participants with a mean disagreement of more than 32 pixels; see S1 Text, Section G).

As a second quality control step, for each participant, we computed the percentage of samples that fall outside a valid AoI, and excluded from the analysis participants for whom this metric was 1.5 IQR above the third quartile (for these participants, more than 70% of all fixations were outside a valid AoI; see S1 Text, Section G). This yielded an eye-tracking sample of N = 33, N = 32 and N = 24 participants in the neutral, positive and negative groups respectively.

We chose this manual quality assurance method because drift can vary significantly between participants and over time. Our manual rater-agreement metric captures this data quality discrepancy, allowing us to formulate a quantitative criterion for excluding participants who have excessive drift (S1 Text, Section G). For participants for whom drift was moderate,

we used custom correction to correct for drift while preserving the relative distribution of fixations across AoIs.

**Relative looking time.** Previous work has shown that the relative looking time to each feature of a multidimensional stimulus is a reliable measure of participants' focus of attention during decision-making [13]. A similar measure was recently used in a study exploring the relationship between uncertainty, choice and eye-gaze [97]. While relative looking time does not account for within-trial dynamics of visual fixations [36–38, 48], it does provide a reliable and interpretable metric of how much relative sensory evidence a participant has collected about one cue versus another. Relative looking time can thus be used to detect biases in value computation.

In our task, relative looking time can be taken as an index of the extent to which participants weighed each of the cues on the display in their decision. This is particularly useful for measuring attentional biases to cues that have higher or lower value, which we hypothesized would be influenced by affect. To compute the relative weight of simple cues within a compound, we summed the durations of fixations to each cue, and divided by the total time spent looking at any of the cue AoIs. This normalization allowed us to interpret relative looking time as a trial-specific measure of attention to each cue.

Because we were only interested in the relative weighting of cues during value computation, we restricted the computation of relative looking time to the time window *before* the participant chose a planet.

## Data quality control

In total, eleven participants (9.2% of the sample) were excluded from all analyses: seven due to failure to learn outcome contingencies (based on a one-tailed binomial test against chance, $\alpha$ = .05), one due to excessive choice of empty stimuli in the learning phase ($> 25\%$), one due to incongruent response (laughter) to the negative affect induction, and two because of computer error resulting in failure to save data. 25 additional participants (20.8% of the sample) were excluded from analysis of eye-tracking data only, as a result of the eye-tracking data quality-control checks described above.

## Statistical analysis

Initial analyses used mixed-effects models as implemented in the lme4 package [98] in R to analyse participants' self-reported mood, choice behavior, and relative looking time. Self-report mood data were analysed using linear mixed-effects analyses, and choice behavior and relative looking time were analysed using mixed-effects logistic regression analyses. Random effects were selected according to a maximal- to minimal-that-converges procedure [99], and incorporated random intercepts for participants as well as per-participant random slopes for effects that were entirely within-participant [100]. *p*-values were calculated using either the Satterthwaite degrees of freedom approximation (for linear mixed-effects analyses; [101]) or the Wald *t*-to-*z* test (for mixed-effects logistic regression analyses).

For relative looking-time data, we weighted each trial by the total duration of fixations within the AOI. As such, trials for which we had more valid eye-tracking data were weighted more heavily, thereby accounting for trial-by-trial differences in the precision of the relative looking time measure. This ensured that, for instance, a trial for which the total fixation duration to cue AoIs was 1000 milliseconds was treated as more informative regarding the latent probability of fixating on different cues than a trial for which the total fixation duration to cue AoIs was 100 milliseconds.

To reflect the hypothesized ordinal relation between affect groups, the video induction was coded as positive = 1, neutral = 0, negative = -1. Compound-generalization block number was recoded as 0, 1, 2, such that coefficient estimates for other predictors reflect the effect of that predictor in the first compound-generalization block, and interactions of that predictor with block number reflect changes in the strength of the predictor's effect relative to the first compound-generalization block. To ensure that the interpretation of the coefficient quantifying the effect of block number on behavior was equivalent across affect-induction groups, the additional baseline generalization block completed by participants in the positive/negative affect groups was not included in analyses of block-dependent changes. Covariates (i.e. per-participant sum scores for each self-report measure) were centered and scaled across participants prior to inclusion in models. For analyses of post-choice mining response times, response times were log-transformed. Trials with mining response times in excess of 2.5 seconds (less than 1 percent of all trials) were excluded from reaction time analyses.

All statistical analyses reported in this manuscript can be reproduced using code and data available in the project's online repository (https://osf.io/egw5c/).

## Computational modelling procedure

To formally test competing hypotheses about the potential effects of affect on compound generalization, we formulated and fit a number of computational models to participants' choice behavior in the compound-generalization phase of the task. These models shared a common choice architecture, which we describe below in its maximal form (i.e., including all parameters). All models tested were variants of this maximal model.

We fit models to data within a hierarchical Bayesian framework, using Hamiltonian Monte Carlo as implemented in Stan [102] to sample from the joint posterior distribution over all model parameters. Four separate chains with randomized start values each took 2750 samples from the posterior distribution. The first 1500 samples from each chain were used for tuning of algorithm hyperparameters ('warmup' phase) and were discarded prior to analysis, leaving a total of 5000 post-warmup samples from the joint posterior distribution for analysis. $\hat{R}$ for all parameters was less than 1.1, indicating acceptable convergence between chains, and there were no divergent transitions in any chain. To optimize sampling speed, all participant-level parameters were drawn from group-level Gaussian distributions whose means and standard deviations were estimated from the data, and then transformed to their respective non-centered parameterization (i.e., all parameters were sampled separately from a unit normal before being transformed to the appropriate range). In particular, to prevent negative values, several parameters were transformed using the normal cumulative distribution function (CDF) to lie in a positive range (softmax $\beta$: standard normal CDF $\times$ 50; decay parameter $\lambda$ and compound bonus parameter $\phi$: standard normal CDF $\times$ 10).

Model comparison was performed using the Widely Applicable Information Criterion (WAIC; [103]), a statistic for comparing models fit with hierarchical Bayesian methods. The WAIC selects models according to their goodness-of-fit to data minus a penalty for the model's effective complexity (estimated as variance of log-likelihood across posterior samples; [104]). We calculated the difference in WAIC between all models and the best-fitting model, and regarded the best-fitting model as credibly better than its competitors if twice the standard error of this difference did not overlap with zero. Ties were broken by selecting the model with fewer parameters (as a coarse proxy for model complexity).

The model recovery analysis was performed by simulating 50 datasets of 100 participants each for each of the four primary models under consideration. For each simulated dataset, we then determined which of the four models provided the best fit to the data under the model

selection criteria described above. The proportion of datasets that were correctly identified as being generated by the true generative model provided an index of the overall identifiability of the models under consideration. The parameter recovery analysis was performed by simulating 100 'participants' for each of the four primary models under consideration, with different parameter settings for each simulated 'participant'. For each model, we then quantified parameter recoverability as the Spearman correlation between actual and estimated parameter values for each free parameter.

**Overview of models.**   We denote the value of a stimulus (i.e., a planet) by $V$ and the value of a cue (i.e., a rune image within a planet) by $v$. The reward-value $v$ of cues was defined as the true generative reward probability for each cue (i.e., 0.25, 0.5, or 0.75) under the assumption that cue values were learned correctly during the learning phase. For simple stimuli (those comprising only a single cue), the value of the stimulus was identical to the value of its constituent cue:

$$V(\text{simple}) = v_{\text{cue}} \tag{3}$$

By contrast, the value of a compound stimulus was assumed to be a weighted linear sum of the values of the two cues comprising it:

$$V(\text{compound}) = \phi \cdot \sum_{j=1}^{2} A_j \cdot v_j \tag{4}$$

where $A_j$ is the attention weight assigned to cue $j$ and $\phi$ is a scaling parameter that captures the general tendency for compound stimuli to be treated as more valuable ($\phi > 1$) or less valuable ($\phi < 1$) than simple stimuli. In this case, $\phi = 1$ corresponds to weighted averaging of simple cue values (as predicted by configural theories of compound generalization; e.g., [54]), whereas $\phi = 2$ (combined with equal attention to both cues, $A_1 = A_2 = 0.5$) would better correspond to summation of simple cue values, as predicted by elemental theories of compound generalization (e.g., [53]).

Within a compound, the attention weight $A_j$ of cue $j$ was proportional to the reward-value of cue $j$ (weighted according to the $\alpha_V$ parameter), and attention weights were then normalised across cues to ensure that the sum of attention weights for all cues within a single stimulus was 1:

$$A_j = \frac{\exp(\alpha_V \cdot v_j)}{\sum_{k=1}^{N} \exp(\alpha_V \cdot v_k)} \tag{5}$$

Here, $\alpha_V$ is a parameter that controls the degree to which attention is biased towards higher-value compounds (when $\alpha_V > 0$, attention is drawn to higher-value cues; when $\alpha_V < 0$, attention is drawn to lower-value cues). We also considered an additional set of models in which attention might be biased towards or away from cues as a function of the certainty with which they predicted different outcomes [105, 106]. However, model recovery analyses indicated that the attention-to-certainty component of these models was not identifiable on the basis of data from this task, and we therefore do not consider this possibility further here.

Lastly, we assumed that choices were generated by a softmax function:

$$Pr(a = i) = \frac{\exp(\beta \cdot V(i))}{\sum_{x=1}^{2} \exp(\beta \cdot V(x))} \tag{6}$$

where $i$ denotes the chosen stimulus, $X$ is the set of available stimuli, and $\beta$ is an inverse temperature parameter that controls the stochasticity of choices.

We selected the best-fitting computational model using a two-step procedure: in step 1, we identified the model that provided the best fit to compound-generalization choices in the absence of an affect induction (i.e., all compound-generalization choices for participants in the neutral affect group, and choices from the baseline block for participants in the positive- and negative-affect groups). In step 2, we tested which parameters of the best-fitting model were modulated by the affect induction itself (step 2 models were fit to all trials in the compound-generalization phase from all affect-induction groups, including the neutral group).

For each Step 2 model, we estimated two variants: one in which the effect of affect on parameters was constant across the three compound-generalization blocks, and one in which the effect of affect weakened between blocks according to a power-law decay function. For an affect-related change in $\alpha_V$, for instance, the effective parameter value in a given block was computed as

$$\alpha_V(\text{effective}) = \alpha_V + \Delta\alpha_V \times \lambda^{\text{block}} \times \text{Affect} \qquad (7)$$

Here, $\lambda$ is a block-wise decay parameter; when $\lambda = 1$, there is no decay in the strength of affect-related parameter changes from one block to the next; by contrast, when $\lambda = 0$, Affect *only* influences parameters in the first compound-generalization block (blocks were zero-indexed, and so $0^0 = 1$) and has no effect in later blocks. In Eq 7, the free parameter $\Delta\alpha_V$ therefore quantifies the affect-related modulation of $\alpha_V$ in the first compound-generalization block. A single $\lambda$ parameter was applied equally to all affect-related changes in each parameter, and $\lambda$ was estimated as a subject-level draw from a single group-level Gaussian distribution with a mean and standard deviation estimated from the data.

## Citation diversity statement

Recent work in several fields of science has identified a bias in citation practices such that papers from women and other minority scholars are under-cited relative to the number of such papers in the field [107–115]. Here we sought to proactively consider choosing references that reflect the diversity of the field in thought, form of contribution, gender, race, ethnicity, and other factors. First, we obtained the predicted gender of the first and last author of each reference by using databases that store the probability of a first name being carried by a woman [111, 116]. By this measure (and excluding self-citations to the first and last authors of our current paper), our references contain 11.66% woman(first)/woman(last), 21.97% man/woman, 22.68% woman/man, and 43.7% man/man. This method is limited in that a) names, pronouns, and social media profiles used to construct the databases may not, in every case, be indicative of gender identity and b) it cannot account for intersex, non-binary, or transgender people. Second, we obtained predicted racial/ethnic category of the first and last author of each reference by databases that store the probability of a first and last name being carried by an author of color [117, 118]. By this measure (and excluding self-citations), our references contain 4.44% author of color(first)/author of color(last), 14.48% white author/author of color, 15.93% author of color/white author, and 65.16% white author/white author. This method is limited in that a) names and Florida Voter Data to make the predictions may not be indicative of racial/ethnic identity, and b) it cannot account for Indigenous and mixed-race authors, or those who may face differential biases due to the ambiguous racialization or ethnicization of their names. We look forward to future work that could help us to better understand how to support equitable practices in science.

## Supporting information

**S1 Text. Supporting information.** All supporting information can be found within the associated PDF file.
(PDF)

## Author Contributions

**Conceptualization:** Daniel Bennett, Angela Radulescu, Yael Niv.

**Data curation:** Daniel Bennett, Angela Radulescu, Sam Zorowitz, Valkyrie Felso.

**Formal analysis:** Daniel Bennett, Angela Radulescu.

**Investigation:** Daniel Bennett, Angela Radulescu.

**Methodology:** Daniel Bennett, Angela Radulescu, Sam Zorowitz, Valkyrie Felso.

**Project administration:** Daniel Bennett, Angela Radulescu.

**Resources:** Daniel Bennett, Angela Radulescu, Yael Niv.

**Software:** Daniel Bennett, Angela Radulescu, Sam Zorowitz.

**Supervision:** Yael Niv.

**Writing – original draft:** Daniel Bennett, Angela Radulescu.

**Writing – review & editing:** Daniel Bennett, Angela Radulescu, Sam Zorowitz, Valkyrie Felso, Yael Niv.

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
