## [Decision Letter · Decision Letter 0]

13 Mar 2023

Dear Dr Bennett,

Thank you very much for submitting your manuscript "Affect-congruent attention modulates generalized reward expectations" for consideration at PLOS Computational Biology.

As with all papers reviewed by the journal, your manuscript was reviewed by members of the editorial board and by several independent reviewers. In light of the reviews (below this email), we would like to invite the resubmission of a significantly-revised version that takes into account the reviewers' comments. As you can see from the reviews, all reviewers liked the study and thought it would make a significant contribution to the field. However, they has several reservations, which we would like to see addressed in a revised manuscript

We cannot make any decision about publication until we have seen the revised manuscript and your response to the reviewers' comments. Your revised manuscript is also likely to be sent to reviewers for further evaluation.

Sincerely,

Tobias U Hauser, PhD

Academic Editor

PLOS Computational Biology

Marieke van Vugt

Section Editor

PLOS Computational Biology

Reviewer's Responses to Questions

**Comments to the Authors:**

Reviewer #1: This article investigates the effects of affect inductions on reward-based choice via an attentional mechanism. The authors induce negative or positive affect using short videos, then examine behavior on a a generalization task with ambiguously valued stimuli. They find that positive affect leads to more optimistic valuation of the ambiguous compound stimuli, and that this change is accompanied by a shift in looking time to the high value stimuli, at least for one of the conditions. The authors also employ a simple model to estimate attention from the choice data. Here they find that attention is modulated by affect.

Overall I found the article to be quite interesting and well written. I appreciated the mix of affective and cognitive psychology, along with eye-tracking and computational modeling. It is unfortunate that there isn’t better agreement between the modeling and the eye-tracking analysis (though see my comments below), but I don’t think that this should be held against the authors. The experiments were well designed and the hypotheses made perfect sense. It just turns out that the story may be more complex than expected. If the authors can address the few questions/concerns that I have, I would be happy to endorse publication.

(1) The authors focus their analysis on choices between L/H compound stimuli and either M or M/M stimuli. This is obviously where they expect to see the biggest effect of their affect manipulation on choice. However, shouldn’t we also expect to see differences in the M vs. L/M and M vs. M/H trials as well, either in choices or in reaction times? Positive affect participants should be slower in M vs. L/M and faster in M vs. M/H compared to negative affect participants. Similarly, positive affect participants should be faster than negative affect participants in L/M vs. L/H and negative affect participants should be faster than positive affect participants in L/H vs. M/H. There should be no effect of affect on L/M vs. M/H.

It also wasn’t clear which analyses were run on the probe trials vs. the non-probe trials. Were the eye-tracking analyses run on all trials or just the probe trials? If it is the former, perhaps this explains why there is a discrepancy between those results and the modeling results, which seem to have been run on all the data.

(2) Regarding the modeling - I had a couple of related concerns. The finding that delta_phi was largely negative has me concerned about a tradeoff between that parameter and the alpha parameter. They appear to be pulling in opposite directions, suggesting that the overall change might be negligible. This made me wonder why the authors chose the specific functional form for the attention weights in equation 2. Why not instead simply estimate weights A_H, A_M, and A_L in each condition? This would eliminate equation 2 altogether and any assumptions about the functional form of attention. With only three value levels this should barely increase model complexity.

(3) Regarding the eye-tracking - I didn’t understand the need for manual analysis here. Do the results substantially change without this drift correction? Or without this weighting by total fixation time? The EyeLink 1000 Plus should not require such corrections, but perhaps it was necessary in this case. If so, more explanation would be helpful.

Minor comments:

(4) I thought that the opening example with the diner was ill chosen, in particular the notion that a high price would lead to a lower reward expectation. We know that high prices actually lead people to expect, and experience, higher rewards - see Hilke Plassmann’s fMRI research on wine tasting.

(5) Ideally Figure 2F would be done for the three conditions separately, perhaps using colored dots here as well?

(6) Table 2 - please indicate how standard errors were computed for delta-WAIC. The table also makes reference to a “Model 8”. I think the authors mean Model 4?

(7) On page 11 it would be helpful to include the equation for the decay lambda, otherwise one has to flip over to the Methods section to be able to interpret it.

(8) The authors might find it helpful to reference work on attribute-based attention: Fisher 2021 in OBHDP; Yang and Krajbich 2022 in Psychological Review.

Reviewer #2: This is a great paper — very clearly written, well-motivated, and beneficial both to basic and applied work on modeling affect, its relation to decision making, and what the two might reveal about latent mechanisms of psychopathology. I hope my comments below help in making final refinements to this paper and the larger project of understanding mechanisms of affect.

“For example, if our diner were to attend more to positively valenced stimulus features (i.e., features that are predictive of reward, such as a preferred style of cuisine), they would form a higher reward expectation for the restaurant than if they attended more to its negatively valenced features (such as a high price)”

Although the example makes complete sense, it assumes that value in inherent to stimulus features, and not, for example, conditional on goals. Consider a situation where for instance someone was inferring the quality of the restaurant, and high prices were indicative of such quality. Price then becomes positively valenced. In line with succeeding comments in the Discussion on the ecological validity of the current paradigm compared to other popular emotion/RL experiments (because rightfully it takes into account that complex states have multiple features to which one could selectively attend, and are integrated in determining state value), I think it’s useful to take into account that the reward function may shift as goals shift.

Is there variation in comparing high and low mean options? For example, is there evidence that a 75/25 option is treated as worse than 75/50? We would expect this pattern even if people simply attended to the highest-value option — so it would be good to see that when highest-value option is held constant between options, the mean value still matters.

“We conducted a model-recovery analysis to ensure that our experimental design was able to differentiate between these models”

Which models? “These” implies referents that were already stated, but I don’t seem them until after this sentence.

You mention individual difference analyses that were non-significant in the main text. It would be good to know roughly what you probed for and why you did so in the main text, as it may not only be of general interest, but meaningfully relate to statements you make below regarding psychopathology and affect-induced attentional biases.

As a general clarification, it would be good to know how the statement — “We focused our analysis on two kinds of choice trials…” — involving compound and simple trial types speaks to the data that went into all analyses. Does that mean that computational analyses also only were on these subset of trials?

Is the preference for compound stimuli suggestive of the possibility that attention tends to be biased towards the higher-valued option in the compound stimuli? Is there a tradeoff between this parameter and the attention weights? Or does this mean people preferentially choose lower-valued compound stimuli even when it isn’t possible for the value to be higher than the simple stimulus regardless of biased attention? For example, do you have evidence to show that the preference for compound stimuli occurs at a greater rate beyond chance that individuals choose a 25/50 stimulus over a single 75 stimulus? Or even 25/50 over a 50 stimulus? I’m not even sure if the modeling incorporated these trials, but they may be useful in comparing your winning model to an alternative that simply decides preferentially based off of either the maximum or minimum value in the compound’s pair of features. I don’t believe this latter model was tested (this could be a third form of feature integration besides averaging or summing over them).

This point relates to a question I had about the robustness of the modeling. Did you, in addition to model comparison, endeavor to recover the parameters of your model? This would certainly help answer whether in fact there is tradeoff in the preference for compound over simple stimuli and the attentional weighting parameters.

Continuing with interpreting the bias towards compound stimuli; does this preference suggest aversion to uncertainty was at play in your paradigm? If so, it might this limit the generalizability of your findings to other situations in which affect-induced attentional biases emerge. Subject ostensibly knew simple stimuli in key comparisons were 50/50, whereas all other stimuli had less outcome entropy (even if their average integration would yield the same level of outcome entropy). From my understanding, in the model, the effect of positive and negative emotion induction is equivalent but in opposing direction (which you tested thoroughly by comparing to other ways in which the inductions may have asymmetrical effects). Might this not necessarily be the case if you de-confound preference for compound stimulus and uncertainty aversion? It could, for example, reveal that the effects of of a negative affect induction is stronger than a positive affect induction, given that the preference towards the simple stimulus is now less uncertain (where that preference I believe indicates attention to negative features in the competing compound stimulus). I guess one way to implement this would be to focus on manipulating reward magnitude as opposed to probability.

Was there any evidence that subjects forgot the values of stimuli over the course phase 2 of the experiment, and specifically, whether subjects selectively forgot stimulus values conditional on the affect induction? I suppose it’s possible that an affect induction could alter how we retain information in memory, which then goes on to predict what we attend to. I think an assumption of the current model is that one’s current affect influences attention at the time of value formation. Is there a way to rule out that individuals merely were less certain about the affect-induction-incongruent stimuli, and thus didn’t take those stimuli into account when making subsequent decisions (reflect in choice preference and eye gaze)?

Your work sheds new light on a prior finding that those who worry tend to perseverate on punishment avoidance goals (Sharp, Russek et al. 2022). There, subjects reporting high anxiety tended to incorporate learning about punishment avoidance features on trials where subjects were instructed to ignore such features. This would suggest that potentially attentional biases to threat, engendered by an anxious emotion, may override instructions to modulate attention. I think it may be worth mentioning this connection, given the potential of your model to extend to RL paradigms requiring individuals not only to attend to multiple features, but to explicitly not attend to certain features given instructed or learned goals. This could bolster your discussion on the ecological validity of your model — in the real world, not only do we need to attend to multiple features, but also alter how we prioritize stimulus features conditional on our shifting goals. That all said, I realize the conflict introduced by my suggesting this work (even if I did not de-anonymize myself, it is implied reviewers may part of suggested literature to cite), and so, I would like to reassure the authors that engaging this point here or in text won’t influence my judgment of their work in the least.

Figure S9 there’s a reference to a Table that reads, “Table ??”

Last, I was surprised but inspired by the inclusion of the “citation diversity statement” at the article’s conclusion. It’s heartening to know this problem is not only being taken into account when considering citations, but also that the practice is being explicitly broadcast as a teaching signal to us all so that we can actually try to improve the current situation. Besides reading a thoughtful paper that advances science I intrinsically enjoy, I’m fortunate to have been selected as a reviewer of this paper to be exposed to this practice. I look forward to implementing it in my future work.

If anything in the above requires further clarification, please don’t hesitate to reach out.

Dr. Paul B Sharp

paul.sharp@mail.huji.ac.il

Reviewer #3: Review of Bennett et al. ‘Affect-congruent attention modulates generalized reward expectations'

Summary

In this interesting study, the authors examine how positive and negative moods bias the evaluation of compound stimuli, whose value depends on a combination of attributes. The aim is to test a mechanistic account for a well-known finding (and intuition) that mood biases evaluation of events.

To do so, the authors use a task wherein participants choose between cues whose associated reward probabilities were learned previously, whilst also undergoing eye tracking. The authors focus on choices between medium-reward cues (either singly or in pairs) and pairs of high- and low-reward cues (H-L pairs).

The key behavioural findings are i) preference for the H-L pair depends on induced-mood, being higher following positive mood induction and lower following negative mood induction ii) relative time spent looking at the high probability cue is associated with higher choice of the H-L pair, and iii) for a subset of choices, mood induction increases the relative time spent looking at the mood-congruent cue.

The authors fit a computational model wherein compound stimuli are valued as a weighted average of their constituent cue values. Through model comparison they show that models in which the weighting parameter transiently changes following mood induction outperform alternatives without such modulation by mood induction. The authors interpret the weighting parameter as reflecting allocation of attention, and support this interpretation with reference to the eye-gaze findings. They suggest that this offers a mechanism by which mood modulates attention paid to mood-congruent features of events.

Major Comments

I found the study to be well thought-out and the manuscript well written. I appreciated the clear figures, which linked up nicely with the main text. The model fitting methods and statistical analyses appeared appropriate to test the hypothesis and were overall clearly described. Overall I thought the paper was impressive in its execution.

My only significant concerns relate to the interpretation of the results:

1) The authors interpret their computational model in terms of relative allocation of attention. However they also acknowledge that a weighted contribution of cues could be interpreted in terms of other cognitive process. I think it could be emphasised more that it is difficult to disentangle effects of ‘attention’ from integration of value. For instance, it seems likely that gaze and value integration are co-dependent. (The authors do allude to this in the Discussion, with reference to sampling-based models but it could perhaps be foregrounded more).

2) The authors state that their eye gaze findings suggest mood-dependent deployment of attention to the high vs low probability cues in compound stimuli. However they compute gaze-time relative to all other cues on the screen, rather than relative to the other cue in the high-low compound stimulus. The former metric (relative to all other cues) would increase with time spent looking at the high-low compound stimulus as a whole. In other words, if a subject were to spend more time looking at the high-low compound, apportioned equally between high and low cues, by the authors’ metric, the proportion of time spent looking at the high cue would increase.

Thus, the authors cannot safely conclude that mood affects the allocation of attention to high vs low components of the compound stimuli themselves. This is important because relative attention assigned to H and L cues within the compound underpins the analogy to the computational model. As presented currently, it seems the findings could be explained if attention paid to the high-low compound - as a whole – varied following mood induction. This would be consistent with any model wherein mood increased the value of the high-low compound.

Also, with respect to the above, the results are described in a potentially misleading way. For instance on p15 the authors write (my italics):

“We next asked whether the general tendency to look at the low- or high-value cue within an L-H compound drives changes in choice. We used a mixed-effects analysis to test whether, in all probe trials, time looking at the high-value cue within an L-H compound stimulus predicted choice of that stimulus, controlling for main effects of affect condition. Indeed, we found that the more time people spent looking at the high-value

cue relative to time looking at all other cues in the trial, the more likely they were to choose the L-H stimulus over an M or M-M stimulus of the same expected value…”

Here, “within an L-H compound” suggests time spent looking at the H cue relative to the L cue (i.e., H/(H+L)), but this isn’t the metric that is tested. The authors should at least include the result of an analysis based on gaze time expressed as H/(H+L), even if this is non-significant (e.g. in a supporting figure).

3) The authors link eye gaze findings with choice proportions (% choose L-H). However, they don’t test whether eye gaze is related to the key model parameter, alpha_v. This missing link would close the loop of the argument nicely. As things stand, the connection with the model is by analogy only. This perhaps makes the model fitting exercise appear a little redundant.

Minor Comments

Figure 1 –very nice, but the cues appear quite small – not sure if there’s a way to make them a bit larger or more contrasting, for display purposes?

P3 “positive, neutral or negative, between participants” On first reading I missed the fact the direction of mood induction was a between-subjects manipulation. The between subject aspect is mentioned only in brackets here and could perhaps be emphasised more.

P5-6 Several results are given as beta coefficient estimates with p-values – it’s not clear without referring to the Methods what kind of regression each refers to. The authors might for instance briefly state the type of analysis, at the start of each section e.g. mixed effects, between subjects etc. Also, later in the Results section the authors state F stats and partial eta squared, while here no test statistics are given.

P7 - The wording when describing weighted averages is unclear –

“Based on previous studies of compound generalization, and given

no explicit instruction on how to estimate the values of these new stimuli, we assumed that participants would estimate the value of a compound stimulus as a weighted average of the values of the cues within the compound (Rescorla, 1976; McLaren & Mackintosh, 2002, one alternative to which is a non-linear conjunction rule; c.f. Saavedra, 1975; Holland, 1991). A qualitative prediction of such a linear rule is that the value of a compound stimulus should be proportional to the mean of the values of its constituent cues..”

It seems misleading that “such a linear rule” follows the previous clause, which mentions a “non-linear rule”.

P7 – “A qualitative prediction of such a linear rule is that the value of a compound stimulus should be proportional to the mean of the values of its constituent cues”.

Doesn't this prediction encounter a limit as the weighting parameter for any one of the cues (A_j) approaches 1?

P& “These findings support the explanation that participants estimated compound cue values using a linear weighted average”.

I’m not sure I follow this. Since the task only examines compounds involving up to two cues, how can non-linear and linear weightings be distinguished? Is not the case that either model boils down to a single parameter (A_1, as expressed below in my comment with reference to the model definition)?

P8 “There was no evidence for an association between any individual-difference measure and performance in either probe trial type, and no interaction between any individual difference measure and the strength of the effect of the affect induction on choice behavior (all p > .10)”

Perhaps there could be a reference to supporting material or Methods here for a list of individual difference measures tested.

P9 Model definition. The model is presented in general form – however, for compounds of two cues it reduces to:

V(compound) = A_1*v_1 + A_2*v_2

Where A_2=1-A_1.

Figure 3

What is the scale for the horizontal axis for the alpha_v parameter here? Equation 2 suggests that alpha_v can be unbounded, yet the authors place a vertical dotted line at alpha_v=0, suggesting that the plot shows alpha_v for the high valued cue relative to the low value cue. The description of the results on p11 also suggests this relative scaling of alpha_v.

Figure 4

This shows decay over time as predicted by the model. It would be nice to see how the model fits to the observed data. For instance could the authors promote Figures S8a and S8b to the main text, and show model fits to the observed pattern of change across blocks?

Figure 5

Here the plotted markers for the medium reward cue are rather faint.

Figure 5c

The authors show fitted lines for each mood induction group, that suggest an interaction. However, they don't refer to analysis of a [group x looking-time] interaction in the Main Text. Incidentally I’m not convinced about the interaction, which seems to rest on one data point, but would be worth stating if this was tested or not.

In the paragraph on p15 referring to Figure 5c, the authors state a chi-squared statistic with only one-degree of freedom – is this correct? And why is a chi-squared test used here?

p 15 The authors asked “whether the general tendency to look at the low- or high-value cue within an L-H compound drives changes in choice”. However, from the analysis we can’t really establish a causative relationship between gaze and choice, since gaze time may also be a function of the value integration process.

Figures 5c vs. 6

The authors could more clearly draw out, in their description of these two figures and associated analyses, whether comparisons are between- or within-subjects. For instance in Figure 5c is there a single data point per subject?

P16 The authors refer here to ‘simple probe trials’. However earlier in the Results section a terminology such as ‘simple vs simple’ and ‘simple vs compound’ was used. On reading the Methods it appears these are different ways of referring different kinds of probe trial – a consistent terminology would make things clearer for the reader.

Discussion

P17 “Taken together, these results suggest a potential computational and perceptual-level mechanism for the well-documented finding that positive affect is associated with optimistic future expectations”

The results do outline a putative mechanism– namely selective attention to affect congruent stimulus features. With a particularly critical reading, it could be argued that the model is simply an alternative description of “the well-documented finding that positive affect is associated with optimistic future expectations”, especially given concerns over the interpretation of the eye-gaze data. Overall however, I think the study is interesting, and that the formalism lends itself to further investigation of how mood might bias attention to different stimulus attributes in a valence-dependent way. The conclusions might be strengthened by examining a link between relative gaze time and the attention weights derived from the model.

**Have the authors made all data and (if applicable) computational code underlying the findings in their manuscript fully available?**

Reviewer #1: Yes

Reviewer #2: Yes

Reviewer #3: Yes

PLOS authors have the option to publish the peer review history of their article (what does this mean?). If published, this will include your full peer review and any attached files.

Reviewer #1: No

Reviewer #2: **Yes: **Paul B. Sharp

Reviewer #3: No
---

## [Decision Letter · Decision Letter 1]

23 Nov 2023

Dear Mr. Bennett,

We are pleased to inform you that your manuscript 'Affect-congruent attention modulates generalized reward expectations' has been provisionally accepted for publication in PLOS Computational Biology.

As you can see, reviewer 1 and 3 made some minor suggestions that I would kindly ask you to implement at the proof stage of the paper (adding analysis to supplement, correcting typos).

Best regards,

Tobias U Hauser, PhD

Academic Editor

PLOS Computational Biology

Marieke van Vugt

Section Editor

PLOS Computational Biology

Reviewer's Responses to Questions

**Comments to the Authors:**

Reviewer #1: The authors have addressed my concerns. I would have liked to see the additional analyses from R.1.1. added to the supplementary material, but I leave this up to the editor to decide.

Reviewer #2: The authors have addressed well all my comments and questions.

Reviewer #3: The authors have thoroughly addressed my concerns with additional analyses, clarifications and additions to the Discussion.

Overall, the results provide clear support for the hypothesis that mood induction biases the valuation of compound stimuli, which is formalised by the computational model. The results also provide partial support for the hypothesis that mood biases visual attention in a mood-congruent manner. I recommend publication of this interesting and well-executed paper.

My only substantive comment is that, as the authors point out, the paper is now quite long, partly as a result of the various clarifications added. Perhaps this could be helped by moving some of the details of model comparison to the supplement. I would not insist on this however, and will leave at the editor's discretion.

Two other very minor points:

1) The authors added a footnote, prompted by my comment, that the weighted average of attention weights can be expressed more simply for the case of two stimuli. In this footnote the authors write the average as alpha*v1 + (1-alpha)*v2. Strictly speaking, shouldn't this be A1*v1 + A2*v2, where A2=1-A1? The authors' use of alpha here makes for a little confusion because it's clear from later modelling analyses that alpha is unbounded, while A is bounded between 0 and 1.

2) There's a small typo in the Discussion section 3.3, "ccalled ambiguity".

**Have the authors made all data and (if applicable) computational code underlying the findings in their manuscript fully available?**

Reviewer #1: Yes

Reviewer #2: None

Reviewer #3: Yes

PLOS authors have the option to publish the peer review history of their article (what does this mean?). If published, this will include your full peer review and any attached files.

Reviewer #1: No

Reviewer #2: **Yes: **Paul B Sharp

Reviewer #3: No

---

## [Editor Report · Acceptance letter]

18 Dec 2023

PCOMPBIOL-D-23-00214R1 

Affect-congruent attention modulates generalized reward expectations

Dear Dr Bennett,

I am pleased to inform you that your manuscript has been formally accepted for publication in PLOS Computational Biology. Your manuscript is now with our production department and you will be notified of the publication date in due course.

With kind regards,

Anita Estes
